# The inheritance of a Mesozoic landscape in western Scandinavia

Ola Fredin[1,2], Giulio Viola[1,3,†], Horst Zwingmann[4], Ronald Sørlie[5], Marco Brönner[1,6], Jan-Erik Lie[5], Else Margrethe Grandal[5], Axel Müller[7,8], Annina Margreth[1], Christoph Vogt[9] & Jochen Knies[1,10]

*In-situ* weathered bedrock, saprolite, is locally found in Scandinavia, where it is commonly thought to represent pre-Pleistocene weathering possibly associated with landscape formation. The age of weathering, however, remains loosely constrained, which has an impact on existing geological and landscape evolution models and morphotectonic correlations. Here we provide new geochronological evidence that some of the low-altitude basement landforms on- and offshore southwestern Scandinavia are a rejuvenated geomorphological relic from Mesozoic times. K-Ar dating of authigenic, syn-weathering illite from saprolitic remnants constrains original basement exposure in the Late Triassic ($221.3 \pm 7.0$–$206.2 \pm 4.2$ Ma) through deep weathering in a warm climate and subsequent partial mobilization of the saprolitic mantle into the overlying sediment cascade system. The data support the bulk geomorphological development of west Scandinavia coastal basement rocks during the Mesozoic and later, long-lasting relative tectonic stability. Pleistocene glaciations played an additional geomorphological role, selectively stripping the landscape from the Mesozoic overburden and carving glacial landforms down to Plio–Pleistocene times. Saprolite K-Ar dating offers unprecedented possibilities to study past weathering and landscape evolution processes.

[1] Geological Survey of Norway, Leiv Eirikssons Vei 39, 7491 Trondheim, Norway. [2] Department of Geography, Norwegian University of Science and Technology, 7491 Trondheim, Norway. [3] Department of Geology and Mineral Resources Engineering, Norwegian University of Science and Technology, 7491 Trondheim, Norway. [4] Department of Geology and Mineralogy, Kyoto University, Kitashirakawa Oiwake-cho, 606-8502 Kyoto, Japan. [5] Lundin Petroleum AS, 1366 Lysaker, Norway. [6] Department of Petroleum Engineering and Applied Geophysics, Norwegian University of Science and Technology, 7491 Trondheim, Norway. [7] Natural History Museum, University of Oslo, 0318 Oslo, Norway. [8] Natural History Museum, London SW7 5BD, UK. [9] ZEKAM/FB5 Geowissenschaften, University of Bremen, 28334 Bremen, Germany. [10] CAGE - Centre for Arctic Gas Hydrate, Environment and Climate, University of Tromsø, 9037 Tromsø, Norway. † Present address: Department of Biological, Geological and Environmental Sciences - BiGeA, University of Bologna, 40126 Bologna, Italy (G.V.). Correspondence and requests for materials should be addressed to O.F. (email: ola.fredin@ngu.no).

Deeply weathered crystalline basement rocks are common features of the geological record and are currently actively developing in warm and temperate climates. When weathering is effective for long periods in tectonically stable regions, it will affect the local geomorphology through pervasive saprolitization, relief reduction and possible development of planation surfaces[1,2]. In repeatedly glaciated areas such as Scandinavia, however, where glaciers and ice sheets have recently stripped large volumes of rocks, saprolites are only sparsely found. The limited saprolite occurrences that still do exist have therefore been considered as crucial archives of information about local pre-Pleistocene climatic conditions and landforms[3–7].

Some of the Scandinavian saprolites, predominantly of the so-called grussy type, are considered to be of Plio-Pleistocene age, as they are developed in glacially sculpted terrain[5,8,9] or are constrained in time by stable isotope temperature proxy data[10]. Recent studies have shown that grussy saprolite indeed can form rapidly mainly through biotite oxidation and associated mechanical break-up of the rock column, which is consistent with a Plio-Pleistocene origin[11]. Other saprolite outcrops, particularly in southern Scandinavia, however, appear significantly more clayey and are thought to represent more mature weathering, and are therefore considered to be older than the last glaciations[5,12,13]. This is additionally supported by stratigraphic constraints at a few key localities that clearly show that, for example, some southern Swedish (Ivö site) and Danish (Bornholm site) saprolites are undoubtedly of Mesozoic origin, although their exact age remains elusive[5,6,12–14].

With uncertainties remaining as to the timing of the weathering episodes that shaped the regional morphology, the origin and age of landscapes in formerly glaciated Scandinavia is thus still controversially discussed[15–25]. The current debate revolves mainly around the morphotectonic history of the Scandinavian mountains, whereas the low-relief terrain close to sea level, although as striking and peculiar, has received less attention. This particular terrain commonly exhibits numerous hills and lake basins overprinted by small-scale glacial erosion landforms and is generally referred to as a landscape of areal scour[26]. It is thought to form through glacial erosion, although the magnitude of the removed rock column is unknown[26–28]. Other conceptual models argue that the landscape of areal scour, for example that in Scandinavia, results from processes typical of cratonic areas dominated by bedrock weathering and stripping as typically found in the tropics, and that its morphology reflects only limited glacial overprinting[4,29,30].

A pertinent example of this scientific debate is provided by the discussion centred on the 'strandflat' morphology of coastal Norway, which was identified already more than a century ago by prominent explorers such as Hans Henrik Reusch and Fridtjof Nansen as a distinct, yet elusive landscape[31,32]. The strandflat is a flat coastal basement terrain, < 60 km wide, with a relief generally between 20 m b.sl. and 50 m a.sl. This landscape type is mainly found along large portions of the Norwegian coast[33], with limited occurrences along the coasts of Svalbard, Scotland, Arctic Canada and the Antarctic peninsula. Based on this geographical distribution, most studies argue for a Pleistocene origin through a combination of glacial erosion, frost shattering, sea ice erosion and wave abrasion[33,34]. However, again based on morphological similarites with deeply weathered and stripped bedrock terrain, some suggest that the strandflat might be a rejuvenated Mesozoic etch surface that has been exhumed and re-exposed through late Neogene erosion[35,36]. The origin and age of this landscape thus still remain open for debate.

This study provides a solution to this problem by dating weathering of the strandflat landscape at Bømlo by K-Ar isotopic analysis of illite clay separated from saprolitic remnants genetically connected to strandflat basement weathering processes. Our new geochronological results indicate a Late Triassic (~ 210 Ma) age for the saprolitization of exposed basement rock in the coastal, western Scandinavian landscape, thus requiring deep weathering and probable landscape formation at that time.

## Results

**K-Ar dating of authigenic illite**. Here we report the first results of K-Ar dating of authigenic, syn-weathering illite separated from three saprolitic outcrops in southwestern Scandinavia (Fig. 1a). Attempts to date supergene (authigenic) minerals as a proxy for weathering and hence landscape forming episodes have been made in the past by K-Ar and $^{40}Ar/^{39}Ar$ geochronology[37–43]. The results from those studies, however, have remained equivocal and the methodology has never been tested in formerly glaciated terrains where saprolite samples are sparse. It has been suggested that single crystal $^{40}Ar/^{39}Ar$ dating of manganese oxides best constrains the timing of continental weathering and geomorphological evolution, because very small and precisely picked samples can be used[39]. $^{40}Ar/^{39}Ar$ dating, however, suffers from well-known drawbacks when applied to extremely fine-grained material such as authigenic clays (< 10 μm or finer) due to $^{39}Ar$ recoil during sample irradiation[44]. To circumvent these problems, we relied on recent methodological and conceptual advances that have made K-Ar dating of clay a valuable tool to date brittle fault rocks[39,44–50]. The separation, characterization and dating of illite in multiple grain-size fractions, where the age invariably decreases with grain size, allows to constrain the role of increased authigenesis in the finest fractions and assign an age to the episode of weathering that contributed to the crystallization of the finest illite clay population[51]. A detailed account of laboratory procedures is found in the Methods section.

**Description of the studied sites**. To test this analytical approach, we sampled two saprolitic sites where there exists reliable and independent stratigraphic control on the age of saprolitization (Ivö and Utsira; Fig. 1a,b,d). In addition, we also investigated unconstrained saprolite samples from the strandflat landscape in coastal western Norway (Bømlo; Fig. 1a,c). The kaolinitic Ivö site in southern Sweden is developed in Mesoproterozoic granitic basement rocks (1.3 Ga old), which experienced prolonged tectonic activity along the Sorgenfrei-Tornquist suture zone during the Mesozoic, with development of horsts and grabens, and an irregular, structurally controlled morphology. The sampled locality is located at the edge of a recently re-activated Cretaceous graben, where a ~ 12 m-thick kaolinite-rich saprolite horizon is located directly on the host granite. The sample, Ivö 1, is from about 20 m above sea level, in the middle of the saprolite profile (Fig. 2a). The saprolite is directly overlain by Early Campanian (Late Cretaceous) sedimentary rocks, which provide a minimum age of saprolitization of ~ 80 Ma[10,30,52].

Two additional saprolite samples were collected at 1,943 and 1,927.5 m b.sl. on the Utsira High horst structure of the North Sea (Fig. 1a,b), from two offshore oil exploration wells (samples Utsira 16/3-4 and 16/1-15, respectively; Figs 1a,b, 3a and 4a). The wells penetrate through ~ 1.8 km of sedimentary rocks (> 1.9 km below current sea level) into weathered granitic rocks of the Utsira High[53]. The basement granite is of Ordovician age (440–480 Ma)[54] and is capped by Late Jurassic (16/3-4) and Early Cretaceous sandstones (16/1-15)[53]. The area experienced volcanism and extension in the Permian, in addition to continued rifting and tectonism from the early Mesozoic onwards[53]. Samples 16/3-4 and 16/1-15 were taken from cores ca. 100 mm in diameter, which offered only limited access to dateable

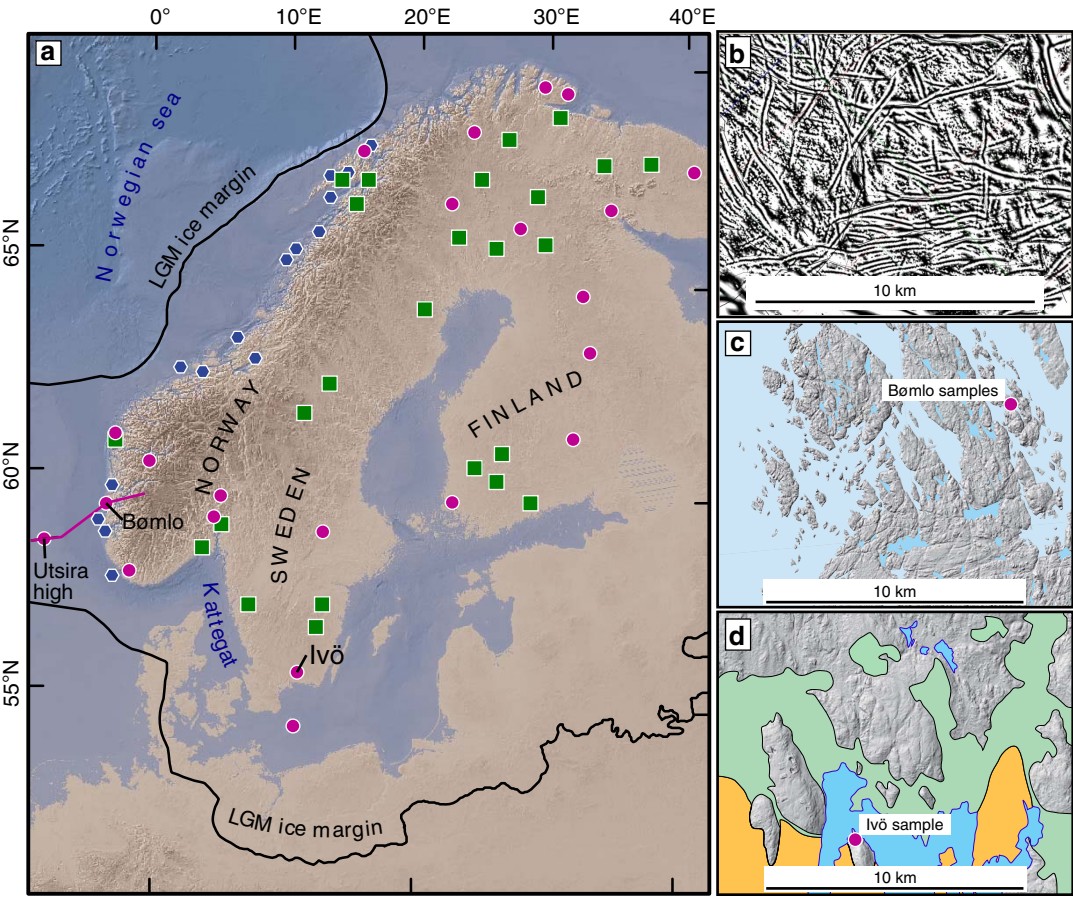

**Figure 1 | Deep weathering and saprolitization in crystalline basement in Scandinavia.** (**a**) Map of known saprolite locations and sample sites (Utsira High, Bømlo and Ivö). Remnants of clay rich (fuchsia circles) and grussy (green squares) saprolites in Scandinavia[5,10,13,84] are shown together with near coast Jurassic basins (blue hexagons)[58]. The black line shows the Weichselian last glacial maximum (LGM) position in formerly glaciated Scandinavia[86]. The fuchsia line stretching from the Shetland Platform, across Utsira High, Bømlo to south central Norway refers to the location of the profile in Fig. 6. (**b**) Seismic image showing the fractured and etched top crystalline basement of the Utsira High beneath almost 2 km of Mesozoic and Cenozoic sedimentary cover. (**c**) LiDAR-derived hillshade map of the Bømlo strandflat in western Norway, exhibiting fractured and weathered crystalline rocks. (**d**) LiDAR-derived hillshade map of Ivö area showing fractured and weathered crystalline rocks protruding through Cretaceous (orange) and Pleistocene (green) cover[30,52].

material (Figs 3a and 4a). In both wells, the upper basement rocks are strongly fractured and altered. The bulk rock is moderately to highly weathered and exhibits alteration of primary minerals to mainly chlorite, illite, smectite and kaolinite clays[53]. The Utsira High site has experienced sedimentation and subsidence since the Late Jurassic, with a significant acceleration during the Pleistocene with vast volumes of glacially derived sediments being deposited on the continental shelf[23,53,55].

Three more saprolite samples were analysed from the island of Bømlo, SW Norway, the onland correlative of the Utsira High (Fig. 1a,c), which is characterized as a strandflat landscape[33] (Fig. 5a–d). Bømlo has a similar geological history to the Utsira High site, except that it might have been eroded during the late Neogene and Pleistocene when the basement was likely stripped from the overlying Mesozoic strata[56]. A few pockets of preserved saprolite overlay a granodioritic host rock of mid-Palaeozoic age (466 ± 3 Ma)[57]. Three near-shore marine Jurassic outlier basins have been mapped close to Bømlo (Bjorøy, Utsira and Karmsundet basins), indicating that Mesozoic strata cover the basement rocks closely offshore the sampling site (Fig. 1a)[58]. The landscape is presently characterized by bare bedrock with negligible Pleistocene cover and with visible joints and faults that form local basins reflecting a long and complicated brittle fault history[57]. Samples Bømlo 2, Bømlo 3 and Bømlo 4 all

come from the same outcrop, an ∼5 m-wide saprolitic corridor bound by poorly exposed subvertical fractures within otherwise generally fresh granodiorite (Fig. 5a–c). Although samples Bømlo 3 and 4 are from the centre of the outcrop and are mature saprolites, Bømlo 2 was collected right at the host rock/saprolite interface and is a texturally and mineralogically less mature saprock (Fig. 5c).

**K-Ar dating results**. The new K-Ar data display progressively younger ages with smaller grain size for all dated samples (Table 1 and Fig. 6a), which is interpreted as reflecting a mixing curve between coarser protolithic or detrital illite/muscovite and fine-grained authigenic illite[49,51]. Illite can be a soil-forming or saprolite clay mineral in weathered rocks, where it forms mainly by alteration of K-feldspar or biotite. It is commonly found in association with other clay minerals such as smectite, when expressing the early phases of the weathering process, or kaolinite during more advanced stages of saprolitization[59]. Illite presence in saprolites reflects a combined process involving both mechanical comminution of (old) protolithic phyllosilicates during weathering and concurrent synweathering authigenic crystallization, where illite incorporates the potassium leached from altering K-rich mineral phases. When authigenesis occurs

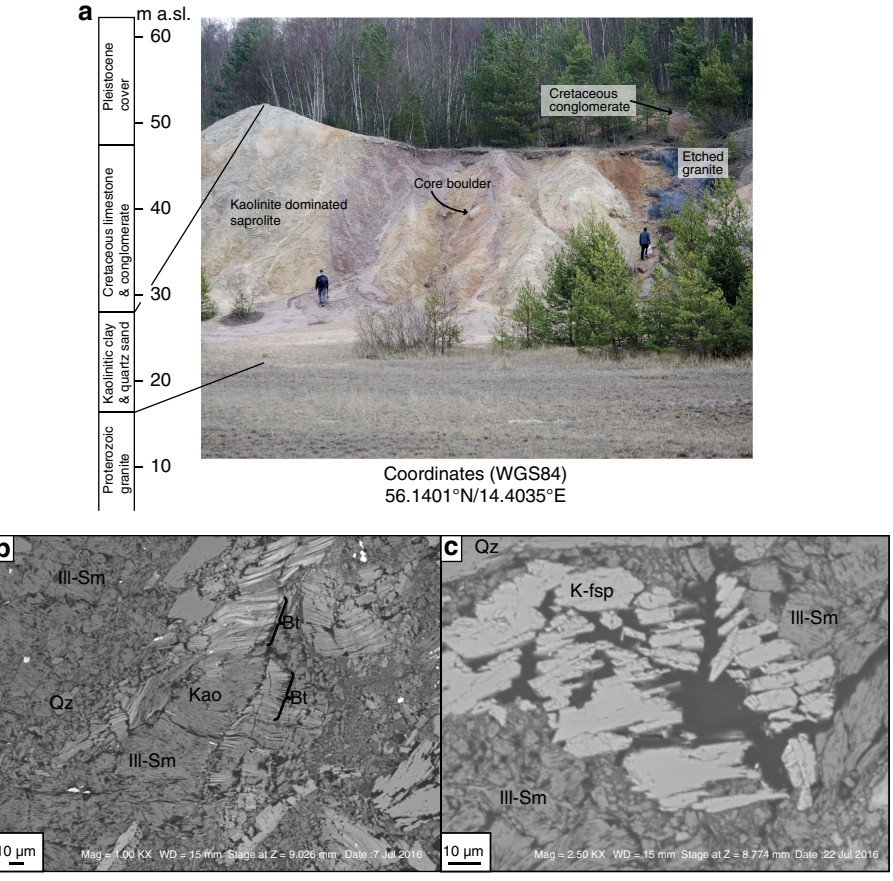

**Figure 2 | Characterization of deeply weathered basement in southern Sweden.** (**a**) Simplifed log and photograph of the sampled saprolite at the Ivö site. (**b**) SEM image from saprolite thin section showing the main clay mineralogy and weathered biotite (Ill, illite; Sm, smectite; Bt, biotite; Qz, quartz; Kao, kaolin). (**c**) SEM image from saprolite thin section showing illite-smectite mixed-layer clay forming at the expense of K-feldspar (K-fsp).

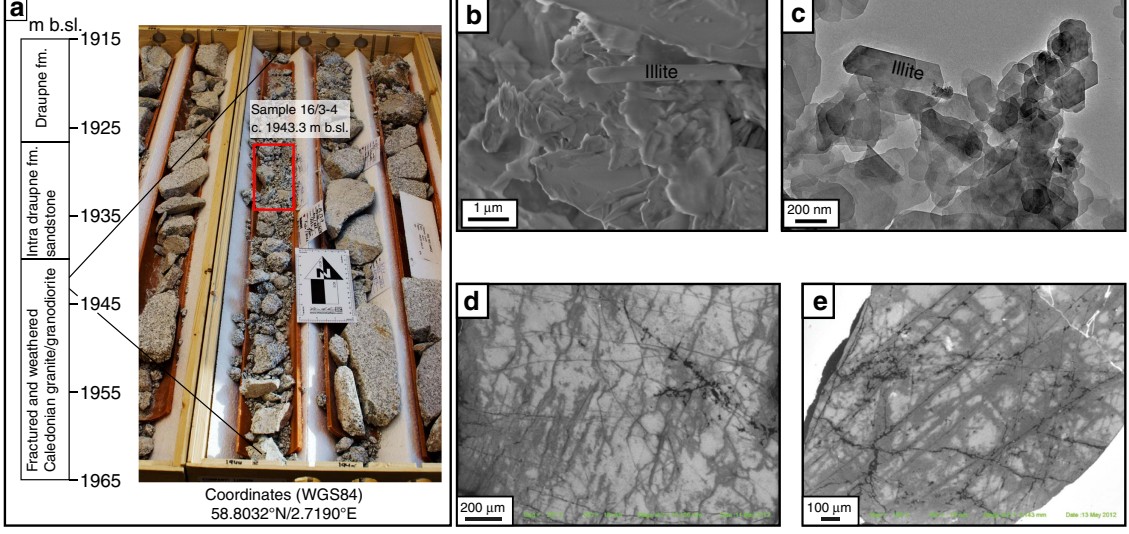

**Figure 3 | Characterization of deeply weathered basement on Utsira High.** (**a**) Simplified log and sample location within well core 16/3-4. (**b**) SEM image of illite from sample 16/3-4. (**c**) Transmission electron microscopy image of illite crystals from sample 16/3-4. (**d**) SEM-CL image of quartz from sample 16/3-4. (**e**) SEM-CL image of quartz from overlying Draupne Fm. sandstone. The images in **d**, **e** are very similar and suggest that the Draupne sandstone was derived locally from weathered basement.

at shallow and relatively cold conditions, illite crystallizes preferentially with the 1M polytype[59]. Illite might also form at higher temperatures, where the dominant illite crystallization polytype can be 2M$_1$. During a retrograde evolution, illite exhibits a tendency to transform into vermiculite or interstratified illite/smectite[59,60].

The model accounting for the possibility to date illite in saprolite is conceptually transferred from the current

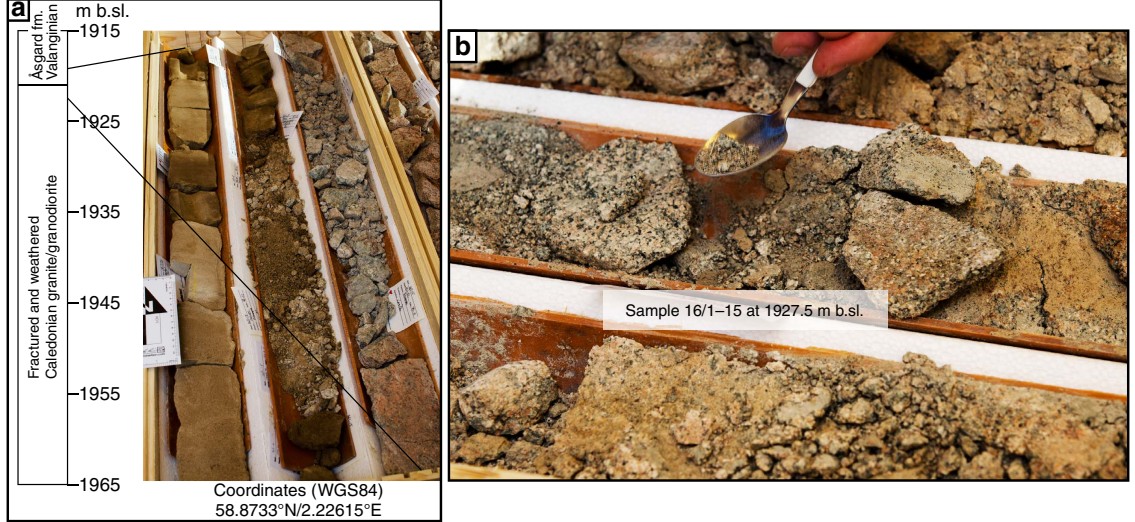

**Figure 4 | Characterization of deeply weathered basement on Utsira High.** (**a**) Simplified log and transition from weathered basement to Åsgard Fm. in well core 16/1-15. (**b**) Sample 16/1-15 at 1927.5 m b.sl. in well core 16/1-15.

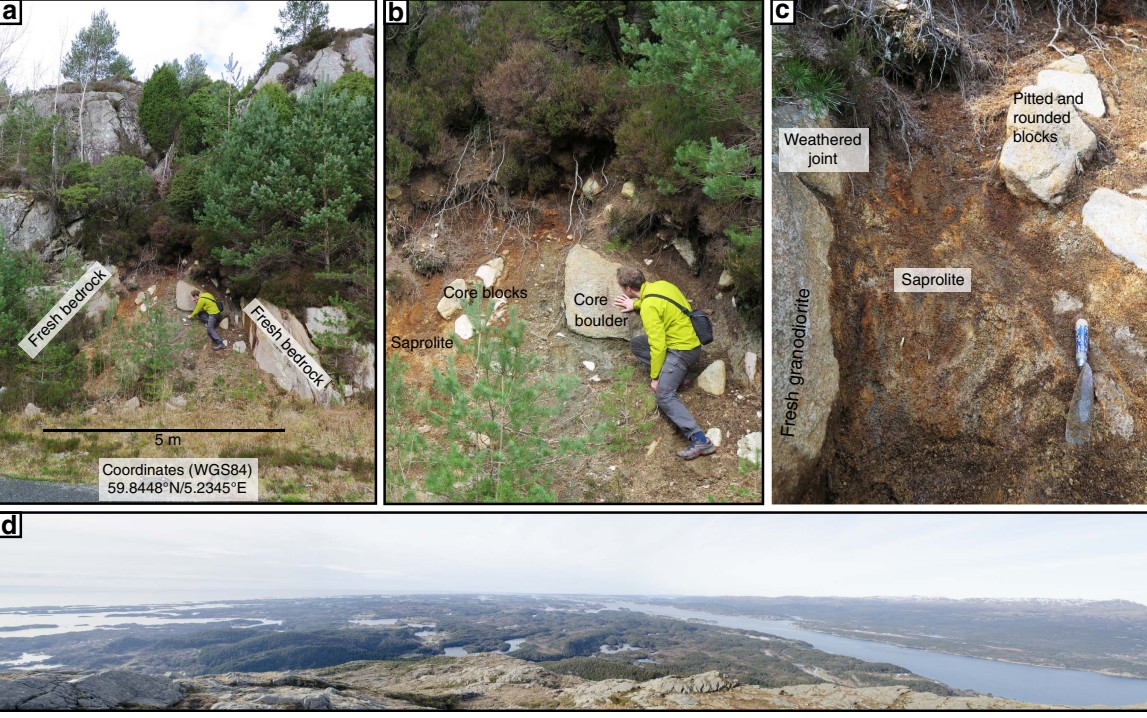

**Figure 5 | Exposed deeply weathered basement on Bomlø in southwestern Norway.** (**a**) Sample site on Bømlo. (**b**) Saprolite and core boulders at sample site, samples Bømlo 3 and 4 are from central portion of the outcrop. (**c**) Detailed view of leftmost part of the outcrop. Sample 2 was taken close to the fresh granodiorite to the left. Spatula is 200 mm long. (**d**) Panorama from mount Siggjo (474 m a.s.l.) looking north towards the strandflat landscape of Bømlo.

understanding of illite synkinematic authigenic growth in brittle faults[45,48,49]. In summary, the adapted model predicts that for inclined age spectra, the age of the finest fraction is representative of the age of authigenesis and neoformation of illite during alteration in connection with deep weathering and development of a saprolitic mantle.

The model can be applied to the Utsira High and the age spectrum of sample Utsira 16/3-4. The coarsest fractions (2–6 μm) yields an age >320 Ma, reflecting a significant protolithic component from the host Ordovician granite (440–480 Ma). The intermediate grain size (<2 μm) shows a younger K-Ar age (231.3 ± 4.7 Ma) and is interpreted as reflecting a mixed assemblage of authigenic syn-weathering illite and detrital illite/mica (Table 2). In the finest available grain size fraction (<0.4 μm), a Late Triasssic (Rhaetian) age of 206.2 ± 4.2 Ma was obtained (Fig. 6a). This is regarded as being close to the actual age of saprolitization and the influence of detrital illite from the protolith is considered irrelevant. This interpretation is supported by detailed clay petrography by scanning electron and transmission electron microscopy investigations, confirming the presence of well-crystallized idiomorphic fibrous illite of authigenic origin in the finest

**Table 1 | K-Ar geochronological data.**

| Sample ID | Grain size fraction (μm) | K (%) | Rad. $^{40}$Ar (mol g$^{-1}$) | Rad. $^{40}$Ar (%) | Age (Ma) | Error (Ma) |
|---|---|---|---|---|---|---|
| Bømlo 2 | <0.1 | 0.15 | 5.6391E − 11 | 60.6 | 210.0 | 13.1 |
| Bømlo 2 | <0.4 | 0.12 | 6.5531E − 11 | 40.3 | 290.2 | 10.6 |
| Bømlo 2 | <2 | 0.17 | 1.0118E − 10 | 12.4 | 307.5 | 38.3 |
| Bømlo 2D | <2 | 0.16 | 8.6333E − 10 | 36.7 | 295.5 | 9.9 |
| Bømlo 2 | 2–6 | 0.24 | 1.8685E − 10 | 17.9 | 406.9 | 20.3 |
| Bømlo 3 | <2 | 0.30 | 1.2097E − 10 | 77.0 | 217.3 | 4.8 |
| Bømlo 4 | <2 | 0.21 | 8.9715E − 11 | 74.3 | 233.0 | 5.0 |
| Ivö 1 | <0.1 | 0.12 | 5.0638E − 11 | 53.6 | 221.3 | 7.0 |
| Ivö 1 | <0.4 | 0.09 | 5.2571E − 11 | 56.5 | 325.4 | 10.3 |
| Ivö 1 | <2 | 0.22 | 1.7339E − 10 | 77.2 | 403.6 | 9.3 |
| Ivö 1 | 2–6 | 0.52 | 4.6804E − 10 | 93.8 | 456.9 | 9.4 |
| Utsira 16/3-4 | <0.4 | 1.32 | 5.002E − 10 | 89.3 | 206.2 | 4.2 |
| Utsira 16/3-4 | <2 | 1.48 | 6.335E − 10 | 89.9 | 231.3 | 4.7 |
| Utsira 16/3-4 | 2–6 | 1.89 | 1.184E − 9 | 97.1 | 329.2 | 6.6 |
| Utsira 16/1-15 | <2 | 3.14 | 1.422E − 9 | 73.5 | 243.9 | 5.0 |

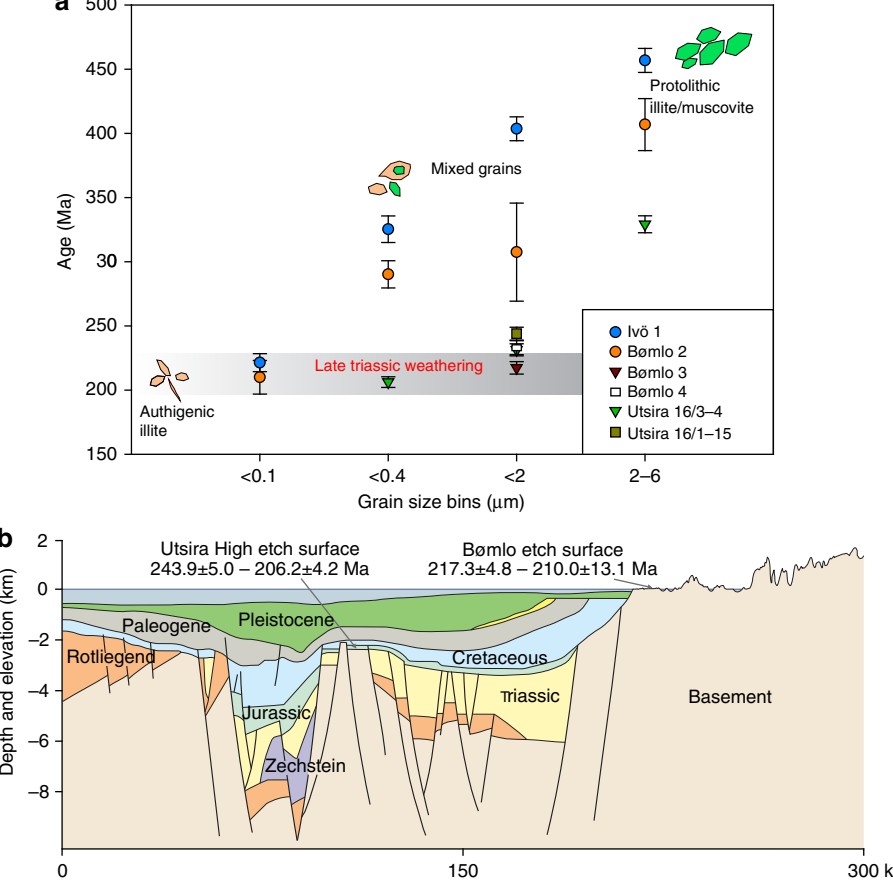

**Figure 6 | K-Ar geochronology of deeply weathered basement in southwestern Scandinavia.** (**a**) Illite K-Ar age (Ma) versus grain size (μm) spectra for the analyzed samples from Ivö, Bomlø and Utsira High. Only the finest grain size fractions are interpreted as representing authigenic illite and are used to infer the age of saprolitization. Error bars are ± 2σ. (**b**) Schematic geological profile along an E–W cross-section across southern Norway and part of the Norwegian Sea[53] (see Fig. 1 for profile location) with the new apparent saprolitization ages. We suggest that the deeply weathered landscapes at Utsira High and Bømlo both formed in the Late Triassic.

fraction of the sample (Fig. 3b,c). Importantly, Mesozoic microfossils were identified from cuttings in well 16/1-15. These microfossils confirm the presence of overlying Mesozoic strata. One finding of *Iraquispora* sp., which is specific for the Rhaetian stage, was made from basement core cuttings. Although this finding remains ambiguous as it is not possible to directly track the source, it could potentially indicate that the Utsira High,

where the well core is situated, was subaerially exposed at that time (Table 3).

Saprolite on Bømlo formed at the expense of granitic host rock similar to the Utsira High. K-Ar dating constrains a saprolitization event of a very similar Late Triassic age and, in addition, sample Bømlo 2 allows a more in-depth analysis of the age versus grain-size relationship, with the dated <0.1 μm fraction. Whereas

**Table 2 | XRD data.**

| Sample ID (µm) | Quartz | Orthoclase/ Microcline | Albite/ Anorthite | Kaolin | Illite/ Mica | Illite/Mica 2M$_1$ | Dioctahedral Smectite | Anatase | Lepidocrocite | Calcite |
|---|---|---|---|---|---|---|---|---|---|---|
| Bømlo 2 <0.1 | | | | | | | 89 | | 11 | |
| Bømlo 2 <0.4 | | | 2 | 4 | | | 78 | | 16 | |
| Bømlo 2 <2 | <1 | | 3 | 7 | | | 83 | | 7 | |
| Bømlo 2 2–6 | <1 | | 3 | 8 | | 6 | 79 | <1 | 4 | |
| Bømlo 3 <2 | <1 | | | 76 | | 1 | 23 | | | |
| Bømlo 3 2–6 | 1 | | | 72 | | 3 | 24 | | | |
| Bømlo 4 <2 | <1 | | 1 | 53 | | | 46 | | | |
| Bømlo 4 2–6 | <1 | | 2 | 53 | | 2 | 43 | | | |
| Utsira 16/3-4 <0.4 | <1 | | | 84 | 16 | | | | | |
| Utsira 16/3-4 <2 | 1 | 5 | 5 | 81 | 7 | | | 1 | | |
| Utsira 16/3-4 2–6 | 4 | 6 | 25 | 54 | 9 | | | 1 | | |
| Utsira 16/3-4 WR | 27 | 18 | 34 | 16 | 3 | | | | | 1 |
| Utsira 16/1-15 | 5 | 6 | 15 | 29 | 16 | | 29 | | | |
| Ivö 1 <0.4 | | | | 92 | 1 | | 7 | | | |
| Ivö 1 <2 | | | | 88 | 4 | | 7 | | | |
| Ivö 1 2–6 | <1 | | | 90 | 5 | | 4 | | | |

XRD, X-ray diffraction.

**Table 3 | Mesozoic microfossils in core cuttings from Utsira High well core 16/1-15.**

| Depth (m b.s.l.) | Microfossil |
|---|---|
| 1917 | *Cerebropollenites mesozoicus* |
| 1926 | *Cerebropollenites mesozoicus* |
| 1929 | *Cerebropollenites mesozoicus* |
| 1932 | *Cerebropollenites mesozoicus* |
| 1945 | *Iraquispora* sp. |

Although *Cerebropollenites mesozoicus* are common throughout the Mesozoicum, *Iraquispora* sp. is strictly restricted to the Rhaetian (Late Triassic).

the intermediate (<0.4 µm) and coarse fractions (6–10 µm) yielded ages between 400 and 290 Ma, thus indicating a significant contribution from protolithic K-bearing phases, the 210 ± 13.1 age of the finest <0.1 µm fraction is identical within error to the Utsira 16/3-4 <0.4 µm fraction (206.2 ± 4.2 Ma; Fig. 6a,b). X-ray diffraction (XRD) analysis of the finest fraction of Bømlo 2 indicates predominant smectite and lepidocrocite (Table 2). SEM analysis confirms the presence of limited amounts of authigenic illite as interlayers in smectite, which is the main K-bearing phase in the <0.1 µm fraction. Illite/mica and kaolinite are instead the most abundant phases in the <0.4 µm fraction of Utsira 16/3-4 (Table 2). Owing to limited sample amount, it was not possible to separate sufficient material to measure an age for the <0.1 µm fraction, but it is important to note that a Mid- to Late Triassic age is constrained by several fractions from four different samples, including the <2 µm fractions of samples Bømlo 3 and 4 with a unique mineralogy containing mainly kaolinite, smectite and minor illite/mica. The plateau of Late Triassic age defined by fractions from <0.1 to <2 µm from four sample is thus taken as robust evidence of illite authigenesis and, indirectly, weathering at that time, at all investigated sites.

Semi-quantitative XRD analysis (Table 2) of the Bømlo samples indicates a significant difference among the samples, although they were collected from the same outcrop. Most significantly, kaolinite increases away from the fresh host rock as the amount of smectite decreases. This trend is interpreted to reflect progressively increasing alteration during saprolitization of the host granodiorite, where illite formed at the expense of K-feldspar, plagioclase and biotite in association with smectite for low weathering degrees and kaolinite for more extreme alteration[59,60]. This trend in the clay mineralogy excludes a possible tectonic or hydrothermal origin of the investigated illite and thus reflects necessarily authigenesis during saprolitization. This observation is reinforced by visual mapping of the saprolite outcrop, where fresh granodiorite gradually disintegrates and exhibits more pronounced small-scale weathering morphology (exfoliation, pitted core stones and rounded joints) towards the rust coloured saprolite in the middle part of the outcrop (Fig. 5a–c).

The K-Ar ages from the clay-rich saprolite in Ivö, S Sweden, essentially follow the same trend of the Bømlo site (Table 1 and Fig. 6a). Whereas older ages in the intermediate fractions (>0.4 µm to 6–10 µm) indicate significant influence of detrital K-bearing phases derived from the local protolith (1.3 Ga old), the finest fraction (221.3 ± 7 Ma) generally matches the age from Utsira and Bømlo, indicating a coeval, Late Triassic episode of saprolitization (Fig. 2a) over much of western Scandinavia. SEM investigations of Ivö samples reveal altered biotite and that illite-smectite mixed layer minerals are formed at the expense of K-feldspars (Fig. 2b,c). The obtained Triassic age is partially confirmed by the local stratigraphy, with Early Campanian (ca. 80 Ma) sediments resting above the saprolite. Additional age constraints and independent control on the K-Ar dating results are provided by studies of nearby basins just offshore Ivö, which document a significant ingress of clay-rich weathering products starting in the Rhaetian[61,62]. These results are fully consistent with initiation of onshore saprolitization in the Late Triassic, as directly dated in this study.

## Discussion

During the Middle to Late Triassic, the northward drift of Pangaea and subsequent formation of the Tethys Ocean gradually changed the pattern of global atmospheric circulation and prevailing climatic conditions[63]. With an increased humidity and high atmospheric $CO_2$ concentration—with a fourfold increase at the end-Triassic mass extinction (ca. 201.4 Ma)[64]— western Scandinavia became subject to intense chemical weathering, as documented by clay assemblages dominated by kaolinite both on- and offshore western Norway and in southern Sweden[20,61,62]. Low-temperature geochronological data show that

**Table 4 | K-Ar age standards and airshot data.**

| Standard | K (%) | Rad. $^{40}$Ar (mol g$^{-1}$) | Rad. $^{40}$Ar (%) | Age (Ma) | Error (Ma) | % Difference from recommended reference age |
|---|---|---|---|---|---|---|
| HD-B1-107 | 7.96 | 3.3558E − 10 | 91.61 | 24.16 | 0.36 | − 0.21 |
| HD-B1-110 | 7.96 | 3.3544E − 10 | 91.93 | 24.15 | 0.34 | − 0.25 |
| HD-B1-121 | 7.96 | 3.3294E − 10 | 90.46 | 23.97 | 0.37 | − 0.99 |
| HD-B1-126 | 7.96 | 3.3718E − 10 | 91.79 | 24.27 | 0.42 | + 0.25 |
| HD-B1-123 | 7.96 | 3.3590E − 10 | 92.07 | 24.18 | 0.37 | − 0.12 |
| LP6-133 | 8.37 | 1.9268E − 09 | 97.37 | 128.07 | 1.86 | + 0.13 |
| LP6-135 | 8.37 | 1.9188E − 09 | 97.59 | 127.55 | 1.90 | − 0.27 |
| LP6-140 | 8.37 | 1.9317E − 09 | 97.23 | 128.38 | 1.96 | + 0.38 |
| LP6-137 | 8.37 | 1.9359E − 09 | 97.64 | 128.65 | 1.91 | + 0.59 |
| **Airshot ID** | $^{40}$Ar/$^{36}$Ar | ± | | | | |
| AS103-AirS-1 | 293.96 | 0.32 | | | | |
| AS106-AirS-2 | 295.47 | 0.28 | | | | |
| AS115-AirS-1 | 295.46 | 0.28 | | | | |
| AS117-AirS-1 | 296.86 | 0.31 | | | | |
| AS122-AirS-1 | 295.26 | 0.23 | | | | |
| AS119-AirS-1 | 296.36 | 0.34 | | | | |
| AS119-AirS-2 | 294.76 | 0.30 | | | | |

basement rocks in coastal and offshore Norway were probably exhumed and exposed to subaerial weathering during the Triassic[55,65–67]. Results from our study confirm the importance of this weathering event, directly recorded by a few preserved pockets of saprolite mantle of Triassic age in western Norway and southern Sweden. The then onshore Late Triassic saprolites were subsequently mobilized by either marine or fluvial erosion during rifting, regional transgressions and/or a more pluvial Jurassic climate. SEM cathodoluminescence (SEM-CL) imaging of quartz grains in offshore Late Jurassic sandstones in well 16/3-4 confirms this model, documenting that the grains were sourced from Late Triassic saprolite (Fig. 3d,e). The upland Late Triassic saprolite was eroded and then redeposited, at lower levels in the landscape, as immature sandstone in the Late Jurassic, proving the efficiency of deeply weathered rocks as a sediment source in a Mesozoic sediment cascade system (Figs 2b and 4).

The geomorphology of a fractured, weathered, etched and stripped basement is often called a joint-aligned valley and consists of fresh bedrock knolls interspersed with valleys containing fractured and weathered bedrock[4,30]. This landscape type (areal scour or cnoc-and-lochan) is common not only in Scandinavia, but also in formerly glaciated Scotland and North America, where it is generally interpreted to be the result of Pleistocene glacial erosion. However, three-dimensional seismic data used to image the basement beneath the offshore sedimentary sequence on the Utsira High[68] (Fig. 1b) indicate that the area around the dated wells exhibits a similar morphology to Bømlo, with a joint-aligned valley landscape (Fig. 1b–d), suggesting that they are possible correlatives. Similar morphologies at Utsira High and Bømlo do not necessarily imply that these landscapes have a common age and origin. However, the new K-Ar geochronological data indicate that the Utsira High and onshore landscapes of Bømlo and Ivö might be of similar Late Triassic age such that a common origin is a plausible hypothesis (Fig. 6b). This relationship was tentatively already pointed out by correlating morphological surfaces onshore with inferred surfaces offshore[69].

The presented new data indicate that important low-altitude basement landforms on- and offshore Scandinavia might be inherited from Mesozoic times, when deep weathering in a warm climate etched the bedrock and weathering products were mobilized into the Mesozoic sediment cascade. Saprolite K-Ar isotopic dating confirms a Late Triassic formation age for weathering products that our analysis strongly suggests to be coeval with the sculpturing of the contentiously discussed coastal strandflat landscape at Bømlo in coastal western Norway (Fig. 6). In turn, this supports the notion that there are old, inherited and variably preserved landscape elements still present in formerly glaciated Scandinavia. These findings require relative tectonic stability of this coastal strandflat region since formation more than 200 Ma ago, which has been predicted also in other morphtectonic studies[15]. If the strandflat at Bømlo had been significantly uplifted, it would have been obliterated by intensified erosion occurring far above the base level. If the landscape, on the other hand, had been very deeply buried in the Mesozoic and Cenozoic, diagenetic alteration products would have formed and potentially been preserved within the now re-exposed saprolite and elsewhere. Recent low-temperature thermochronology data also support this understanding by constraining cooling and exhumation in the Late Triassic and Jurassic, thereafter followed by slow burial and heating to no more than 50 °C, which implies Cretaceous burial down to a depth of ca. 1 km[55,65,66]. Cenozoic exhumation then followed to re-expose the strandflat landscape and subject it to recent surface processes and in particular Pleistocene glacial-, periglacial- and marine erosion. However, despite Pleistocene re-exposure, the strandflat gross morphology still resembles a warm climate etch surface[35,36].

Finding Mesozoic saprolites significantly predating the Pleistocene glaciations in formerly glaciated Scandinavia warrants some discussion, because it is often assumed that glacial erosion can efficiently remove all pre-glacial landforms and deposits cutting deep into bedrock. There is, however, mounting evidence that glacial erosion is very heterogeneous, whereby glacial plucking can erode in the order of 2 km of rock in deep fjord basins, while delicate pre-glacial landforms are still preserved in-between glacial troughs. The pattern of glacial erosion and preservation is generally governed by subglacial temperature, which is a function of bedrock topography, ice sheet dynamics, position of the ice divide, geothermal heat flow and atmospheric temperature. It is thus possible that the Mesozoic saprolites escaped the Pleistocene glacial erosion in the same way delicate landforms have been shown to survive several glacial cycles[23,70–73]. Another explanation for finding saprolite outcrops in Scandinavia is that they represent the current (interglacial) erosion level, and that these outcrops might be obliterated after the next glaciation. Instead, after the next glacial cycle (perhaps > 100 ka into the future), new saprolite outcrops might

be exposed after glacial removal of overlying strata close to the current near-shore Mesozoic basins.

In summary, we present an innovative, new approach by applying K-Ar geochronology to authigenic illite clay genetically associated with deep weathering (saprolitization). This approach demands judicious geological field mapping of saprolite outcrops together with clay mineralogical characterization to avoid ambiguous results. If authigenic illite formation is taken as a proxy for deep weathering and there are remnants of saprolite to be found in the landscape, the method opens a new avenue to date ancient weathering and possibly landscape-forming episodes. Our results show that the strandflat landscape at Bømlo and potentially large regions of southwestern Scandinavia were subject to intense deep weathering in Late Triassic time (ca. 220–200 Ma). We believe that this weathering produced a significant landscape that was covered by sediments in late Mesozoic and Cenozoic times, and that was ultimately exhumed and stripped in the Neogene and Pleistocene.

## Methods

**K-Ar dating of saprolite illite.** Detailed accounts of the conventional K-Ar technique can be found, for example, in refs 74,75. Owing to the hygroscopic nature of clays, special care was taken in the preparation of both K and Ar sample splits. For K analysis by AAS (Varian Spectra AA 50), two splits of $\sim 50$ mg (each) sample material were dried overnight in an oven at 100 °C and reweighed using a Mettler AT20 balance. The sample aliquots were dissolved with HF and $HNO_3$. The measured dry weight was used in the calculation of K concentration. The pooled error of duplicate K determination of all samples and standards is better than 2%.

Ar isotopic determinations were performed using a procedure similar to that described by Bonhomme[76]. For Ar analysis by noble gas spectrometry, sample splits were loaded into clean Mo foil (Goodfellow molybdenum foil, thickness 0.0125 mm, purity 99.9%), weighed and subsequently pre-heated to 80 °C overnight to remove moisture, and reweighted using a Mettler AT20 balance. The measured dry weight was used in the K-Ar age calculation. Samples were stored before loading into the Ar purification line in a desiccator. Once loaded into the Argon line, the samples were pre-heated under vacuum at 80 °C for several hours to reduce the amount of atmospheric Ar adsorbed onto the mineral surfaces during sample handling. Argon was extracted from the separated clay mineral fractions and whole rock splits by fusing samples within a vacuum line serviced by an on-line $^{38}Ar$ spike pipette. The isotopic composition of the spiked Ar was measured with a high sensitivity on-line VG3600 mass spectrometer. The $^{38}Ar$ spike was calibrated against standard biotite GA1550 (ref. 77). Blanks for the extraction line and mass spectrometer were systematically determined and the mass discrimination factor was determined periodically by airshots. Sample material (ca. 15 mg) was required for Argon analyses.

The analytical approach to separate, characterize and date illite from up to 1,000 g of saprolitic material followed the methodology described elsewhere[78]. However, the radiogenic isotope systematics of sedimentary or weathered rocks are complex due to the intimate mixture of minerals of different origins such as detrital phases, potentially from a variety of sources, as well as authigenic minerals. Consequently, it is often difficult to unambiguously interpret measured ages. Special sample preparation techniques involving freeze–thaw disaggregation to avoid overcrushing and extensive size separation to reduce the amount of detrital phases can address these issues[79]. Progressive size reduction down to submicrometre size fractions ($<0.1$ μm) increases the proportion of authigenic clay phases in the clay component and minimizes contamination and suggests that the most reliable isotopic ages for authigenic clay minerals are obtained for the finest size fractions. When possible, up to four clay size fractions were separated ($<0.1$, $<0.4$, $<2$ and 2–6 μm). Characterization of the individual fractions was carried out by semi-quantitative XRD (identification and quantification), SEM and selective transmission electron microscopy on clays. Details of the K-Ar dating approach can be found elsewhere[78]. During the course of the study, nine international standards (five HD-B1 and four LP-6) and seven airshots were analysed. The results are summarized in Table 4. The error for Argon analyses is below 1.00% and the average $^{40}Ar/^{36}Ar$ value of the airshots yielded $295.45 \pm 0.29$. The K-Ar ages were calculated using $^{40}K$ abundance and decay constants recommended by Steiger and Jäger[80]. The pooled error of duplicate K determination on all samples and standards is better than 2%, whereas the error for Ar analyses is below 1%. The age uncertainties take into account the errors during sample weighing, $^{38}Ar/^{36}Ar$ and $^{40}Ar/^{38}Ar$ measurements and K analysis. K-Ar age errors are within $2\sigma$ uncertainties. Ages are reported to the timescale of Gradstein et al.[80]

Results invariably define inclined age versus grain-size spectra (Fig. 6a)[45,48,50,51].

**SEM and cathodoluminescence.** SEM investigations were carried out on carbon-coated thin sections using an LEO 1450VP analytical SEM. SEM was applied to document the micro structure of the investigated granite sample and the occurrence of non-quartz minerals within the sand samples by backscattered electron (BSE) imaging. The composition of minerals was determined with an energy dispersive spectroscopy (EDS) detector from Oxford Instruments attached to the SEM. The applied acceleration voltage and current at the sample surface were 20 kV and $\sim 2$ nA, respectively. BSE images were collected from one scan of 43 s photo speed and a processing resolution of $1,024 \times 768$ pixels and 256 grey levels. The same instrument was used for EDS measurements and BSE imaging of freeze-dried aliquots of the sample collected at Ivö. Bulk material was fixed onto an aluminum stub using a small amount of Crystalbond adhesive (Pelco Prod. Nr. 821-2). Eight, uncoated stub-samples were placed in the SEM sample chamber, which was held at variable pressure ($3 \times 10^{-4}$ Pa) to avoid charging of the samples.

SEM-CL images of quartz were obtained with a Centaurus BS Bialkali CL detector attached to the LEO 1450VP analytical SEM. The applied voltage and current were the same as for the SEM investigations. The Bialkali tube has a CL response range from 300 (violet) to 650 nm (red). The detector sensitivity peaks in the violet spectrum range around 400 nm. SEM-CL images were collected from one scan of 43 s photo speed and a processing resolution of $1,024 \times 768$ pixels and 256 grey levels.

**X-ray diffraction analysis.** The samples were lightly front-pressed onto Si low background sample holders for X-ray diffraction analysis. XRD patterns were recorded with a PANalytical X'Pert Pro Multi-purpose Diffractometer using Fe filtered Co Ka radiation, variable divergence slit, 1° anti-scatter slit and fast X'Celerator Si strip detector. The diffraction patterns were recorded in steps of 0.017° $2\theta$ with a 0.5 s counting time per step and logged to data files for analysis. Quantitative analysis was performed on the XRD data from all bulk samples using the commercial package SIROQUANT from Sietronics Pty Ltd. The results are normalized to 100% and hence do not include estimates of unidentified or amorphous materials.

**Data availability.** The authors declare that the data supporting the findings of this study are available within the paper.

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

## Acknowledgements

The study was funded by a joint industry project conceived between Lundin Norway AS and the Geological Survey of Norway (NGU). Further funding came from joint projects TWIN (Norwegian Petroleum Directorate, NGU) and BASE (Lundin Norway AS, AkerBP AS, Wintershall Norge AS, Maersk Oil Norway AS and NGU). We thank G. Peterson (Geological Survey of Sweden) for access to LiDAR data of the Ivö area. A. Todd and M. Raven, CSIRO, are thanked for technical assistance during the study. I. Throndsen assisted with Mesozoic palynology. E. Larsen (NGU) provided comments on the manuscript. J.K. is supported by the Research Council of Norway (RCN grant 223259). Several geologists from Lundin Norway performed initial mapping of Bømlo. I. Lundqvist (NGU) digitized Fig. 6b. Thorough and constructive reviews by Adrian Hall, David Lundbek Egholm and one anonymous reviewer significantly improved the paper.

## Author contributions

O.F., G.V., M.B. and J.K. conceived the study. H.Z. performed K-Ar, SEM and transmission electron microscopy measurements. R.S. provided well core access and interpretation of well stratigraphy, J.-E.L., E.M.G., O.F., G.V. and J.K. mapped the Bømlo area. J.-E.L. and E.M.G. interpreted offshore seismic data and identified offshore saprolite occurrences. A. Margreth performed SEM analysis and A. Müller contributed with SEM-CL data, while C.V. provided quantitative clay mineralogical data. O.F., G.V., H.Z. and J.K. wrote the manuscript with input from all authors.

## Additional information

**Competing interests:** The authors declare no competing financial interests.

