## [Peer Review File · Nature Communications]

Reviewers' comments:

Reviewer #1 (Remarks to the Author):

Deeply and intensely weathered rocks are widespread on the Earth's surface. Whilst many weathering profiles appear to be forming or changing today, others are clearly relict. Relict saprolites are found buried beneath younger sedimentary rocks and sediments or in contexts where the geochemistry and mineralogy of the saprolite is incompatible with Pleistocene and Holocene climatic regimes. These relict saprolites are important archives of palaeo-environments. Prolonged phases of deep weathering are also often associated in the geological record with major unconformities and with periods of relief reduction and eventual planation (Godard et al., 2001).

Whilst much progress has been made in recent decades in dating major phases of erosion and in constraining erosion rates over the last 0.1 to 1 Myr, absolute dating of major weathering phases remains challenging and imprecise. This paper provides results from K-Ar dating of illite clays in southern Scandinavia that are regarded as products of near-surface weathering. If this technique can be demonstrated to produce reliable results then it represents a major breakthrough in the dating of weathering phases. This is particularly important because weathering may penetrate many tens of metres below the ground surface. In slowly eroding terrain, the secondary minerals produced by weathering often provide the last remaining evidence of the conditions at and close to at the time of weathering. The survival of ancient weathering also provides information on long term erosion rates and of the timing of phases of tectonic stability. The dating method reported in this paper is therefore of very wide potential interest to the earth science community.

My understanding of the line of argument presented in the paper is as follows:

1. Clay-rich saprolites occur onshore in Scandinavia and also beneath Mesozoic sedimentary rocks offshore and also in Skåne. The age or ages of the weathering phases that produced these saprolites are poorly known.
2. The finest fraction of the illite clay found in the saprolites is a product of chemical weathering at the time saprolite formation.
3. K-Ar dating of these fine clays indicates that an important phase of weathering occurred in the Late Triassic. This event is recognised at one location on the Utsira High, one location presently on the strandflat south of Bergen and one location in Skåne.
4. Joint-valley terrain, interpreted as the product of etching of fractured basement, is developed below Late Jurassic rocks on the Utsira High, on the strandflat surface and below Late Cretaceous chalk in Skåne and much more widely in formerly glaciated basement terrain.
5. The Late Triassic age of the weathering clays at one site on the strandflat and the form of the rock surface together indicate that the strandflat is an exhumed Mesozoic landscape.

Based on that understanding, each of the main points in the argument can be critically assessed:

1. The age or ages of the weathering phases that produced these Scandinavian saprolites are indeed poorly known. The point can be made also about other basements terrains, within and outside the limits of glaciation. So the successful application of K-Ar dating to weathering clays would represent a major step forward. The authors should however acknowledge that fracturing, alteration and weathering often interact and that weathering profiles may have a multi-phase development.
2. Clays may form in many ways. Given that all the sites considered here once lay or still lie beneath Mesozoic sediments, it is reasonable to ask if the authors can be certain that the youngest clays are not of diagenetic origin? This matters especially for questions of the origins of the strandflat because diagenesis can occur when groundwater circulates through fractured or porous rocks, including saprolite, far below overlying sediments. In contrast, weathering extends generally only a few tens of metres below a land surface.
3. Whilst K-Ar dating has been applied widely to the problems such as the timing of diagenetic clay formation in mudstones (Burley and Flisch, 1989) and of clay in fault gouges (citations in paper), the technique has not been used much in weathering studies. This may be because of the

many challenges in dating very fine and therefore potentially unstable clay minerals in the weathering environment. The authors deal with the problem of identifying detrital illite by analysing different size fractions by showing that the age of the illite clays is partly a function of the size fraction analysed, with older ages in coarser fractions. It is not clear however why even the finest clay fraction should not be heterogeneous and include detrital illite. The TEM/SEM images show that most illite is authigenic in the finest size fraction but are the authors satisfied that no detrital or diagenetic illite is included in the samples analysed? Even a small amount of detrital illite in the $<0.1 \mu$ fraction would give ages that are too old (Hamilton et al., 1989). This consideration matters since the overlying or nearby sedimentary rocks at each locality are {greater than or equal to}50 Myr younger than the proposed Late Triassic ages of the weathering clays.

In this study, K-Ar ages are available for two sites for the $<0.1 \mu$ size fraction but only for $< 0.4 \mu$ fraction at Utsira and for the $<2 \mu$ for Bømlø 3 and 4. In the text, these ages from different size fractions are reported as similar and mutually supporting. But such is the importance of the size of the clay fraction that the actual ages of the $<0.1 \mu$ size fractions could be 80-100 Myr younger at Utsira and 100-180 Myr younger at Bømlø 3 and 4. The data set is not large enough to dismiss such possibilities.

4. Joint-valley terrain is widely recognised in Scandinavia and elsewhere. But because landscapes look similar does not mean that the landscapes share a common origin and age. Joint valley terrain is a result of long term weathering and erosion (etching) operating in areas of low relief developed in fractured basement rocks. It occurs beneath Mesozoic to Cenozoic unconformities and in Cenozoic landscapes, including basement lowlands stripped of weathering by Pleistocene glaciation. Such terrain is fundamentally structurally controlled and may develop wherever and whenever there is prolonged weathering of a basement lowland. There is also some evidence that some joint valley terrains (etchsurfaces) may develop even without a prior phase of deep weathering through continuous weathering and stripping (Thomas, 1989a; Thomas, 1989b). Joint valley terrain is potentially diachronous and polygenetic and so its form cannot be used to infer age. In the context of this study, it is plausible to have joint valley terrain of Triassic, Cretaceous and Pleistocene age at Utsira, Skåne and Bømlø.

5. The authors do not develop properly the argument that the strandflat is of Mesozoic origin, perhaps for reasons of lack of space. The distribution of the strandflat is not shown on Fig. 1, nor its relation to Triassic sedimentary rocks offshore. No locations are shown of the Jurassic outliers known on the strandflat. In particular, the Bjørøy tunnel outlier of early to middle Oxfordian age (Bøe et al., 2010) is quite close to the Bømlø site. The cross section provided from Utsira to Bømlø and into the mountains does not support inheritance - it shows clearly that the sub-Triassic basement surface has been tilted and then deeply eroded, and a new surface formed close to present sea level. Instead the section suggests that this new surface - the strandflat - is of Neogene or Pleistocene age. Whilst locally the strandflat may include elements of exhumed basement surfaces and associated weathering covers of Mesozoic age, the exposure of these remnants at the present erosional level does not require that the strandflat is mainly an exhumed Mesozoic surface.

Specific points

Line

1 Title. For me, the important message in this paper that is new and exciting is that the finest clay fractions from exhumed saprolites have yielded K-Ar ages from the Triassic. Inheritance of Mesozoic landscapes is widespread on many cratons so that finding is not novel. Neither is the claim that the strandflat is an exhumed Mesozoic surface (Olesen et al., 2013). The authors may wish to change the title to focus on the ground breaking aspects of the paper.

7 & 12 Check capitals

24 Perhaps "mainly thought"? Some saprolite occurrences have been regarded as Pleistocene in age (Sørensen, 1988).

25 Weathering in Fennoscandia has a range of ages.

27 Is this first order geomorphology really the strandflat?

30 Do not the stratigraphic constraints suggest a younger age, before the Late Jurassic or Late Cretaceous?

31 Is it not the age determinations that point to this conclusion?

37 This paragraph is concerned with gross morphology of the Norwegian passive margin whereas the paper is more directed to the strandflat.

38 Perhaps the word controversial is sufficient?

38 Is first-order needed here?

40 Over large areas? Do planation surfaces control the first order geomorphology?

45 Interfluvies are lowered through time and so are dynamic features. It has been argued however that the plateaux are inherited planation surfaces formed before uplift. These planation surfaces are potentially old forms, although slow erosion continues to reshape them.

48 In response to denudational unloading?

50 and shaped by glacial and periglacial processes?

52 dynamic forms of Pleistocene age?

53 Space could be saved here by deleting this sentence.

61 I suspect that neither of these references mentions joint valleys or cnot and lochan.

63 This paragraph should perhaps maintain a focus on the strandflat.

71 The authors may wish here or at line 107 to mention any previous attempts to use this K-Ar method to date weathering or alteration clays.

75 Bømlo lies quite close to a Jurassic outlier.

82 The Campanian falls within the Upper Cretaceous. The sediments at this site are early Campanian and so date from closer to 80 Ma.

84 This paragraph does not inform the reader about the nature of the saprolite at Utsira.

99 Overlie

99 Does the granodiorite have an absolute age determination?

107 This is an important paragraph. This might be the place to point out the novelty of the application of the technique to weathering clays.

116 To assign an age, does not the clay fraction need to be entirely homogeneous and authigenic?

126 This statement on the weathering origin of illite needs a reference? The alternative diagenetic origin of illite could be mentioned.

129 This sentence appears to cover two separate points: illite is unstable and only small amounts of fine illite are needed to date authigenesis.

132-135 This sentence sets out what could be regarded as the most novel and important aspect of the paper and so should appear earlier and in the abstract?

138 Should the word weathered be here?

147 In similar rather than at the expense of?

160 The implication here seems to be that a mixed clay assemblage including authigenic illite is dated. See my comments above on reporting ages from mixed size fractions. Why is the authigenic illite necessarily of weathering origin?

171 Why is a hydrothermal origin excluded? Zonation is common across linear zones of alteration. Alteration zones may also be reactivated.

178 The age of kaolinisation at this site generally has been regarded as ~100 Myr younger (Gilg et al., 2013).

188 The SI indicates that the Bømlo occurrence is not clay-rich nor highly kaolinitic. It is unclear how clay-rich or kaolinitic (or thick) is the Utsira weathering profile? So whilst these occurrences may be the roots of Triassic kaolinitic weathering profiles, this needs to be established.

190 The word abrasion might be omitted.

205 As argued earlier, similarity of form provides only weak support for similarity in age.

206 The concluding paragraph makes a huge leap from a set of dates based on a novel application of the K-Ar dating technique that indicate a late Triassic age for authigenic illite at one site on the strandflat to a bold statement on the age and origin of the entire strandflat and on the post-Caledonian tectonic history of western Norway.

References

- Bøe, R., Fossen, H., Smelror, M., 2010. Mesozoic sediments and structures onshore Norway and in the coastal zone. *Norges Geologiske Undersøkelse Bulletin*, 450, 15-32.
- Burley, S., Flisch, M., 1989. K-Ar geochronology and the timing of detrital I/S clay illitization and authigenic illite precipitation in the Piper and Tartan Fields, Outer Moray Firth, UK North Sea. *Clay Minerals*, 24, 285-315.
- Gilg, H.A., Hall, A.M., Ebert, K., Fallick, A.E., 2013. Cool kaolins in Finland. *Palaeogeography, Palaeoclimatology, Palaeoecology*, 392, 454-462.
- Godard, A., Simon-Coinçon, R., Lagasquie, J.-J., 2001. Planation surfaces in basement terrains. In: A. Godard, J.-J. Lagasquie, Y. Lageat (Eds.), *Basement Regions*. Springer, pp. 9-34.
- Hamilton, P., Kelley, S., Fallick, A.E., 1989. K-Ar dating of illite in hydrocarbon reservoirs. *Clay Minerals*, 24, 215-231.
- Olesen, O., Kierulf, H.P., Brønner, M., Dalsegg, E., Fredin, O., Solbakk, T., 2013. Deep weathering, neotectonics and strandflat formation in Nordland, northern Norway. *Norwegian J. Geol*, 93, 189-213.
- Sørensen, R., 1988. In-situ rock weathering in Vestfold, southeastern Norway. *Geografiska Annaler. Series A. Physical Geography*, 299-308.
- Thomas, M.F., 1989a. The role of etch processes in landform development. I. Etching concepts and their applications. *Zeitschrift für geomorphologie*. NF, 33, 129-142.
- Thomas, M.F., 1989b. The role of etch processes in landform development. II. Etching and the formation of relief. *Zeitschrift für geomorphologie*. NF, 33, 257-274.

Adrian Hall

Reviewer #2 (Remarks to the Author):

Review of "The inheritance of a Mesozoic landscape in western Scandinavia" by Fredin et al.

This study presents new age constraints on saprolites along the coastal areas (the strandflats) of the western Scandinavian mountains. The results suggest that the saprolites studied are exceptionally old, dating back to the late Triassic period (~200 Myr ago). The finding hence indicates long-term stability of the strandflat areas in spite of significant erosion further inland and deposition of sediments offshore during several hundred million years.

Some general comments:

The results presented here are important, and I think the study represents a significant step forward for our understanding of long-term landscape evolution in Scandinavia. It also advances a novel quantitative method based on K-Ar dating. Furthermore, the text is generally well written and I enjoyed reading it.

For these reasons I believe the study will attract significant interest. In my opinion the results presented here deserve to be published in *Nature Communications*.

Yet, there is an urgent need to revise the introduction and the discussion of the results, as it is hampered by misunderstandings of previous work (see below). Once these misunderstandings are corrected, however, I think that the authors can significantly improve the discussion and strengthen their conclusions, but it requires that the authors critically rethink the consequences of their findings. I hope the authors will find my comments useful in this process.

The authors generally discuss their findings against two existing hypotheses for development of the Scandinavian topography. One (the 'peneplain-hypothesis', refs 2, 5, 11-13) assumes that the current topography was formed by a series of late-Cenozoic tectonic events that lifted surfaces from sea level (fluvial base level) to their present elevation (up to more than 2 km). It states that the current topography in Scandinavia is generally young (late-Cenozoic), but may include older

elements that have been buried and later exhumed by late-Cenozoic erosion.

The other hypothesis (the 'Isostasy-Climate-Erosion (ICE) hypothesis' ref 6, of which I am a coauthor) is simpler and suggests that the topography is a long-lived slowly eroding and partly collapsed remnant of the Caledonian Mountain belt, which first formed more than 400 Myr ago. This hypothesis does not call for recent tectonic uplift events but relies on climate sensitivity of surface processes to explain variable sediment flux to the surrounding basins.

The misunderstanding I mentioned above relates to the ICE-hypothesis:

In lines 51-52 it reads: "The main implication of this model is that all landscape elements essentially are of the same Pleistocene age"

This statement is simply wrong and very misleading! It is very important to correct it because it unfortunately complicates the discussion and sends the readers in a wrong direction.

The ICE-hypothesis does indeed suggest that the high plateaus in Scandinavia were affected by glacial/periglacial erosion (which does not seem completely farfetched when considering that many of them likely have a >10 Myr glacial/periglacial history). Importantly, however, this does not mean as stated in line 51-52 that all landscape elements in Norway were formed in the Pleistocene, not even the high plateaus (ref 6 suggest they started forming in the Oligocene) and certainly not the strandflats!

In contrast, the ICE-hypothesis proposes that the first-order topographic structure of western Scandinavia is a remnant of the Caledonian mountain belt, which over time has been gradually destroyed by rifting and long-term climate-sensitive erosion. According to the ICE-hypothesis, the coastal areas in particular, have been more or less stable for hundreds of million years (since rifting), because vertical displacements were dominated by isostatic adjustments causing rock uplift in-land and subsidence off-shore, leaving the coast zones as a relatively stable tipping line (influenced however by variable sea levels and sediments that may have piled up in near the coast). There is a section in ref 6 dedicated to the long-term stability of the coastal areas. I suggest the authors reread this before revising their manuscript.

In short, I think the old saprolite ages reported here are very much in agreement with the ICE-hypothesis.

In contrast, it is much more difficult to reconcile Triassic saprolites with the other hypothesis including Late-Cenozoic uplift of peneplains (refs 2, 5, 11-13). According to this hypothesis the uplifted peneplains must become older with elevation, such that the highest represents the oldest preserved landscape and the first uplift event (see age distribution of surfaces in ref. 12).

I believe that the present study in combination with previous work (Goddfellow 2012, Goodfellow et al. 2014, not cited in the manuscript) on younger saprolites at higher elevation demonstrates that it is the other way around: the oldest (most slowly developing) landscapes are near the coast and the youngest are higher up where erosion processes are more active (which, by the way, the ICE-hypothesis would also predict).

The authors fail to discuss this properly because the misunderstanding about Pleistocene landscapes and the ICE-hypothesis stands in the way. In my opinion, the findings reported here first-of-all demonstrate that post-orogenic mountain belts are long-lived features dominated by overall slow erosion processes and weathering (except for where ice-sheets choose to erode) and this finding is important to publish.

Some more specific comments:

line 27-28: How can a handful of samples from the strandflat area and offshore (that is at and below sea level) deliver "... evidence that first order geomorphology of western Scandinavia is a relict from Mesozoic times"? By first-order geomorphology of western Scandinavia I understand the overall shape of the mountain range, but surely this study is not dating this.

This comment actually points to a general problem in this study: dating weathering is not the same as dating a landscape. Weathering landscapes in particular are known to evolve by slow surface lowering under steady-state geomorphic conditions (see for example Anderson 2002,

Geomorphology 46, p. 35-38). In this case, one can time the weathering but it does not make sense to define an age of the landscape. Assigning ages to bedrock landscapes has been a popular activity in Scandinavia, but I think that it should be done with more caution.

line 29: following on from the last comment, it is not clear how basement exposure and weathering are correlated in these saprolites. Can the authors provide an estimated upper limit to the maximum saprolite thickness, and explain how exposure links up with chemical weathering?

line 32: How do we know that the strandflat landscape "initiated" in the Triassic? This study's results demonstrate that the strandflat experienced weathering then, but how do we know that it had just started?

line 48: consider writing "...slow isostatic crustal uplift..."

line 51: The authors must correct this misunderstanding and consider its implications.

line 63: I do not agree that this study is dating the strandflat landscape. The authors are dating the formation of clay minerals found in the landscape (see point above for lines 27-28), which is not the same as the landscape (see comment above).

line 72: I think this test is a clever approach and the method description is a real strength of the study. But I wonder if the method would also work in younger saprolites that have not yet developed large amounts of secondary clay minerals?

line 98: Can the authors say more about the overlying strata that was stripped in the late Cenozoic? Does the saprolite give any constraint on their thickness (from e.g. compaction)? In order to make any sense (see comment above on age-distribution in ref 12), the peneplain model often has to form a landscape, bury it, and exhume it again without the erosion processes changing the form of the original buried landscape. It is difficult to understand how that works, but off course a strong contrast in erodability between the cover and the buried landscape would help. It would be good to see this discussed more here.

The results section is convincing and very well written.

line 161: I wonder from what criteria the sampling sites were chosen? Is it possible that the Triassic age plateau emerges because only very mature saprolites were included in the study? Many weathering profiles in Scandinavia seem to be less mature (again, see Goodfellow's work). Would they challenge the image of a Triassic age plateau if they were measured? Overall, I think that the study needs to include more info on previous work on weathering profiles in Scandinavia, and a broader discussion on how representative the samples measured here are.

line 200: The "cnoc-and-lochan" landscapes likely form because of regional differences in bedrock fracture density. This difference may influence the pace of weathering and fluvial/glacial erosion in the same way, so it is not unlikely that different surface processes can form the same type of landscape, at least on length scales that can be resolved by seismics.

line 212-214: I strongly disagree with this conclusion, and I cannot see how the authors can arrive at this given the sentence before. At least they should explain it better.

Supplementary info: The additional figures (in particular the photos) are great and very useful. I think that the format of Nature Communications allows for including some of them in the main text.

I hope that the authors are given the opportunity to revise the manuscript, as they present important new results. The results do, however, deserve to be discussed in a more convincing

way. The authors are welcome to contact me, should they have any questions.

David Egholm

Reviewer #3 (Remarks to the Author):

A. Summary of the key results:

An interesting paper. K-Ar geochronologic methods are used to establish the age of pedogenically (authigenic) illite in saprolite/weathered rock during Mesozoic era. These dates are used to argue for a long term development of the standflat landscape in western Scandinavia. They observed a decrease of measured K-Ar age in grain size in two saprolites stratigraphically constrained (Ivö and Utsira) and in one saprolite (Bømlø) not constrained to develop their model for stand flat formation. At first glance, these data make sense and lead to the paper being considered by Nature. The paper is fairly well written and provides a good context for wide multidisciplinary readership. There is one stratigraphic miscue (e.g. line 82, "middle" Cretaceous - See Gradstein et al cited in the SI). I am more focused on the K-Ar systematics than landscape formation.

B. Originality and interest: if not novel, please give references:

The use of K-Ar geochronological data to study landscape development is novel and original. Vasconcelos' work is the only work coming close to this effort. The development of terrestrial climate proxies is one of more difficult aspects of the study of ancient climates. This paper opens another possible tool for developing climate proxies. This paper will potentially attract a wide readership.

C. Data & methodology: validity of approach, quality of data, quality of presentation:

The extraction of authigenic (diagenetic, or even pedogenic) illite from rocks, sediments, and soils having mixed provenance is very difficult work. I can appreciate the painstaking efforts in the areas of STEM and characterization to show the presence of this fine-grained illite (and smectite, if it is illite-smectite, see following point) in the presence of other phases including K-bearing phases in some instances (Utsira). I bring up some points needing some clarification.

For the Bømlø saprolites, it appears the K-bearing phases giving rise to the K-Ar ages are smectite (and albite/anorthite?) in the fine fractions (mica is seen only in the coarse fraction). The finest fraction of this Bømlø2 saprolite gives the lowest age and interpreted as authigenic. Could this smectite be an illite-smectite per XRD? If yes, then how does the presence of I-S condition the interpretation and conclusions for the formation of these phyllosilicate minerals? Still authigenic? Pedogenic? Transformation?

The Ivö saprolite shows kaolinite and small amounts of illite-mica. Is this illite-mica a 1Md polytype mica while the other mica is 2M1 as per Table SI1? The K-Ar ages of the Ivö saprolite are most supportive of their argument. Even so, you have potentially two K-bearing clays (smectite, some K is fixed or is it illite-smectite with more fixed K) and illite/mica (1Md illite?). Are both the smectite and illite-mica authigenic?

For the petrographic work, I follow the argument for authigenic illite based on the data presented including the K-Ar data. It is possible to see mica or weathered mica in kaolin-rich saprolites as evident in the rocks nearby my home. The data that would have cinched this argument would have been to show illite being weathered from mica or K-feldspar in these saprolites (e.g. Elliott et al, 1997, CHEM GEOL - Elsevier just posted a cleaner version of this paper). Illite polytype analyses would have helped. It would have been useful to post the XRD patterns in the SI data.

In other words, I am questioner of the pedogenic/authigenic origin of this illite. The K-Ar ages yield younger age signifying either loss of Ar by weathering a muscovite mica from bedrock or creation the authigenic formation of illite. Can a transformed or weathered mica from bedrock be ruled out? If micas are being weathered to illite, is this weathering congruent in terms of K and Ar behavior in this illite/mica? Is more parent or more daughter lost/gained? I would gather more daughter (Ar rad) is lost in weathering.

D. Appropriate use of statistics and treatment of uncertainties:

The treatment of uncertainties is good for the K-Ar data. The measurement of high percentages of radiogenic argon is impressive for the low K samples from the Bømlø2, Ivö samples. The recording of mole Ar-40 rad/g out to five significant figures tells me a microbalance or high-quality analytical balance was used. No measurement of interlaboratory standard noted for K-Ar data. Not enough information is provided about the semi-quantitative XRD techniques.

E. Conclusions: robustness, validity, reliability

The key to this paper is accepting the interpretation of the petrographic data as well as some more detailed exposition of the illite polytypes in the saprolite matrix materials. The authors could have stressed better the alignment of the measured ages and the stratigraphic constraints.

F. Suggested improvements: experiments, data for possible revision:

I agree with the authors that laser- Ar-Ar might not be useful here. Their approach is the best that anyone can do with the available techniques. Having the XRD data available answers questions about the smectite (is it I-S?). Examination of the feldspars for its weathering to illite and ruling out bedrock sources of illite/mica help this argument tremendously.

G. References: appropriate credit to previous work? OK.

H. Clarity and context: lucidity of abstract/summary, appropriateness of abstract, introduction and conclusions

Comprehensible paper.

Reviewers' comments:

Reviewer #1 (Remarks to the Author):

Deeply and intensely weathered rocks are widespread on the Earth's surface. Whilst many weathering profiles appear to be forming or changing today, others are clearly relict. Relict saprolites are found buried beneath younger sedimentary rocks and sediments or in contexts where the geochemistry and mineralogy of the saprolite is incompatible with Pleistocene and Holocene climatic regimes. These relict saprolites are important archives of palaeo-environments. Prolonged phases of deep weathering are also often associated in the geological record with major unconformities and with periods of relief reduction and eventual planation (Godard et al., 2001).

Whilst much progress has been made in recent decades in dating major phases of erosion and in constraining erosion rates over the last 0.1 to 1 Myr, absolute dating of major weathering phases remains challenging and imprecise. This paper provides results from K-Ar dating of illite clays in southern Scandinavia that are regarded as products of near-surface weathering. If this technique can be demonstrated to produce reliable results then it represents a major breakthrough in the dating of weathering phases. This is particularly important because weathering may penetrate many tens of metres below the ground surface. In slowly eroding terrain, the secondary minerals produced by weathering often provide the last remaining evidence of the conditions at and close to at the time of weathering. The survival of ancient weathering also provides information on long term erosion rates and of the timing of phases of tectonic stability. The dating method reported in this paper is therefore of very wide potential interest to the earth science community.

My understanding of the line of argument presented in the paper is as follows:

1. Clay-rich saprolites occur onshore in Scandinavia and also beneath Mesozoic sedimentary rocks offshore and also in Skåne. The age or ages of the weathering phases that produced these saprolites are poorly known.
2. The finest fraction of the illite clay found in the saprolites is a product of chemical weathering at the time saprolite formation.
3. K-Ar dating of these fine clays indicates that an important phase of weathering occurred in the Late Triassic. This event is recognised at one location on the Utsira High, one location presently on the strandflat south of Bergen and one location in Skåne.
4. Joint-valley terrain, interpreted as the product of etching of fractured basement, is developed below Late Jurassic rocks on the Utsira High, on the strandflat surface and below Late Cretaceous chalk in Skåne and much more widely in formerly glaciated basement terrain.
5. The Late Triassic age of the weathering clays at one site on the strandflat and the form of the rock surface together indicate that the strandflat is an exhumed Mesozoic landscape.

We agree with this concise summary

Based on that understanding, each of the main points in the argument can be critically assessed:

1. The age or ages of the weathering phases that produced these Scandinavian saprolites are indeed poorly known. The point can be made also about other basement terrains, within and outside the limits of glaciation. So the successful application of K-Ar dating to weathering clays would represent a major step forward. The authors should however acknowledge that fracturing, alteration and weathering often interact and that weathering profiles may have a multi-phase development.

We are fully aware of the complexities mentioned by the reviewer, but, at the same time, we have good control on the dated saprolitic outcrops, on the saprolite characteristics and on the genetic and time relationships with brittle faulting in the study area. As a matter of fact, the results presented in the paper stem from a multidisciplinary project, whose main goal is to understand, characterize and date the complex ways brittle deformation, landscape development, weathering and saprolitization interact as part of a delicate balance of geological processes. As part of this scientific approach, we always carefully map and sample saprolites in the field and separate them from possible fault gouges or hydrothermal alteration haloes. We also systematically characterize our samples through a multianalytical approach in the laboratory, determining also the polytypism of the separated illite in all grain size fractions. We know very well that brittle deformation may go hand in hand with deep weathering, and a recent paper (Viola et al. Nature Comm, in press) deals with that aspect in the study area. We have added a couple of sections acknowledging these complexities.

2. Clays may form in many ways. Given that all the sites considered here once lay or still lie beneath Mesozoic sediments, it is reasonable to ask if the authors can be certain that the youngest clays are not of diagenetic origin? This matters especially for questions of the origins of the strandflat because diagenesis can occur when groundwater circulates through fractured or porous rocks, including saprolite, far below overlying sediments. In contrast, weathering extends generally only a few tens of metres below a land surface.

The conceptual approach used here to interpret the obtained results relies on the by now quite well established fact that the age vs. grain size relationship (wherein the finer the dated fraction, the younger the age) reflects the progressively increasing amount of authigenic illite in the finest and of the protolith illite in the coarsest. The very recent Viola et al. (Nature Communications, in press) paper demonstrate this for inclined “age-grain size” spectra derived from brittle fault rocks. In this paper we use the same concept and approach, with the difference that illite authigenesis is not synkinematic (that is, during faulting), but is syn-weathering! This means that the small illite crystallites that we use to constrain the age of weathering are volumetrically more important in the finer fractions, whereas they could not grow to coarser sizes (due to thermodynamic and kinetic constraints) and are therefore absent and do not contribute to the age of the coarse size fractions. This is indeed the great advantage of using a multi-grain size approach, that is, the possibility to tell apart the age contribution of the old, “detrital-protolithic” component from that of the syn-weathering authigenesis. The statistical robustness of our dataset, moreover, wherein many samples point consistently at illite formation in the late Triassic, is also a good indication that the method works and that we are dating true authigenic clay. If, for example, groundwater circulation had caused diagenetic alteration and authigenesis at a different point in time, we would expect a large scatter of the age of the finer fractions..

3. Whilst K-Ar dating has been applied widely to the problems such as the timing of diagenetic clay formation in mudstones (Burley and Flisch, 1989) and of clay in fault gouges (citations in paper), the technique has not been used much in weathering studies. This may be because of the many challenges in dating very fine and therefore potentially unstable clay minerals in the weathering environment.

We like to think that this is in part also due to the fact that until recently the conceptual understanding of illite K-Ar dating was still very incomplete. Our group has spent several years refining the concepts and laboratory routines behind its application to complex brittle faulting histories and it was only because of this progress that we could start to run the same approach to illite grown in

a totally different environment, i.e. that of in situ weathering and alteration (see Viola et al., in press Nat. Comm. for the K-Ar direct dating of feldspar alteration within a dilatant brittle fracture zone resulting from important fluid-rock interaction).

The authors deal with the problem of identifying detrital illite by analysing different size fractions by showing that the age of the illite clays is partly a function of the size fraction analysed, with older ages in coarser fractions. It is not clear however why even the finest clay fraction should not be heterogeneous and include detrital illite.

See one of the earlier comments. Detrital illite will also be present in the finer fractions. However, its content will be insignificant in comparison to the content of the authigenic component.

The TEM/SEM images show that most illite is authigenic in the finest size fraction but are the authors satisfied that no detrital or diagenetic illite is included in the samples analysed? Even a small amount of detrital illite in the $<0.1 \mu$ fraction would give ages that are too old (Hamilton et al., 1989). This consideration matters since the overlying or nearby sedimentary rocks at each locality are {greater than or equal to} 50 Myr younger than the proposed Late Triassic ages of the weathering clays.

Indeed every obtained age can be considered as a mixed age. However, the age of the finest fractions is as close as possible to the age of the actual event that we are trying to date.

In this study, K-Ar ages are available for two sites for the $<0.1 \mu$ size fraction but only for $<0.4 \mu$ fraction at Utsira and for the $<2 \mu$ for Bømlo 3 and 4. In the text, these ages from different size fractions are reported as similar and mutually supporting. But such is the importance of the size of the clay fraction that the actual ages of the $<0.1 \mu$ size fractions could be 80-100 Myr younger at Utsira and 100-180 Myr younger at Bømlo 3 and 4. The data set is not large enough to dismiss such possibilities.

We agree with the reviewer that additional $<0.1 \mu\text{m}$ fractions from the Utsira and Bømlo sites would aid with the interpretation. Unfortunately, it was not always possible to separate the $<0.1 \mu$ fraction due to sample nature and clay characteristics. Thus, the $<0.4 \mu$ ages become obviously maximum ages for the finest fraction. However, recent advances using the same method to date fault gouges where there exists by now a considerable dataset to show that it is not uncommon for the ages to plateau below the $<0.4 \mu$ fraction. The age does generally not become much younger at $<0.1 \mu$ in any of the studies on fault gouges. This might be the case also for saprolites. Moreover, it is probably difficult to obtain a much better age by centrifuging the sample to even finer fractions than $<0.1 \mu$, because at these grain sizes we are approaching the primary nucleus in the clay Oswald ripening process and the age might again be skewed. We believe the numerous fault gouge studies show convincingly that the $<0.1 \mu$ -- $<0.4 \mu$ often yield reliable results.

As for the age of the overlying strata and the apparent mismatch of 50-120 Ma: This is certainly a long period of time. At the Utsira High the saprolite is discordantly overlain by Draupne Fm. Sandstone (c. 150 Ma), however there are reports of limited occurrences of older (Triassic) spores in well core cuttings that can be interpreted as indicating that the Triassic is mostly missing in the stratigraphy (hiatus) but the Triassic weathering of the basement is instead still preserved. This would corroborate the interpretation of our results that late Triassic weathering is indeed a possibility at Utsira High. See also Riber et al. (2015) for a qualitative discussion on possible saprolitization age at Utsira High.

At Ivö the capping rock is of Campanian age (c. 120 Ma younger than the Triassic saprolite). However, the surrounding basins show a significant onset of continental weathering already in the

Rhaetian (i.e., the late Triassic) – see studies by Ahlberg et al. This again leaves us with a long hiatus that requires a protective cover of the saprolite through the Jurassic, then erosion and again deposition of Cretaceous sediments. Indeed it is a complicated scenario, but after oral communications with Karna Lidmar-Bergström (the first to fully describe and appreciate the importance of the Ivö site) it seems to be a possibility that this is the case. She (Lidmar-Bergström) also stressed the onset of deep weathering in Rhaetian as an important marker horizon that coincides with the K-Ar data.

4. Joint-valley terrain is widely recognised in Scandinavia and elsewhere. But because landscapes look similar does not mean that the landscapes share a common origin and age. Joint valley terrain is a result of long term weathering and erosion (etching) operating in areas of low relief developed in fractured basement rocks. It occurs beneath Mesozoic to Cenozoic unconformities and in Cenozoic landscapes, including basement lowlands stripped of weathering by Pleistocene glaciation. Such terrain is fundamentally structurally controlled and may develop wherever and whenever there is prolonged weathering of a basement lowland. There is also some evidence that some joint valley terrains (etch surfaces) may develop even without a prior phase of deep weathering through continuous weathering and stripping (Thomas, 1989a; Thomas, 1989b). Joint valley terrain is potentially diachronous and polygenetic and so its form cannot be used to infer age. In the context of this study, it is plausible to have joint valley terrain of Triassic, Cretaceous and Pleistocene age at Utsira, Skåne and Bømlo.

We fully agree with these observations. Our intent was not to interpolate an age between landscapes with similar morphology. We were, however, intrigued by the apparent similarity of the buried landscape offshore (visible in the seismic image) with onshore joint valley terrain (areal scour). Additionally, the consistent and independent constrain of the new K-Ar ages strengthened the correlation of the etched landscape on land western Norway in the strandflat and that seismically imaged beneath the Mesozoic cover on Utsira High.

5. The authors do not develop properly the argument that the strandflat is of Mesozoic origin, perhaps for reasons of lack of space. The distribution of the strandflat is not shown on Fig. 1, nor its relation to Triassic sedimentary rocks offshore. No locations are shown of the Jurassic outliers known on the strandflat. In particular, the Bjorøy tunnel outlier of early to middle Oxfordian age (Bøe et al., 2010) is quite close to the Bømlo site. The cross section provided from Utsira to Bømlo and into the mountains does not support inheritance - it shows clearly that the sub-Triassic basement surface has been tilted and then deeply eroded, and a new surface formed close to present sea level. Instead the section suggests that this new surface - the strandflat - is of Neogene or Pleistocene age. Whilst locally the strandflat may include elements of exhumed basement surfaces and associated weathering covers of Mesozoic age, the exposure of these remnants at the present erosional level does not require that the strandflat is mainly an exhumed Mesozoic surface.

This is a valid point and we try to develop these concepts better in the updated Discussion. We have added the Mesozoic outliers from Bøe et al to Figure 1a (the map), which indeed shows that Mesozoic (Jurassic) sediments often are found closely outboard of the strandflat. We agree that the strandflat probably is polygenetic, and that significant exhumation and reshaping might have happened in the Pleistocene. However, as Olesen et al. (2013) points out there is a problem to comprehend how erosion and evacuation of vast amounts of rock along the Norwegian coast (where the ice sheets have been thin and with diverging flow) happened if the strandflat was carved out of fresh bedrock. The notion of an old etch surface covered with easily eroded sediments (sub-Cretaceous according to

Lidmar-Bergström) and saprolite helps explain how Pleistocene glaciers could manage to strip the rock column down to- and exhume the strandflat. Once the Mesozoic planation surface was exposed however, Pleistocene ice sheets, waves and rivers certainly have scoured and modified the etch surface, as clearly indicated by glacial whalebacks, P-forms and similar Pleistocene landforms. It seems possible though, that the gross morphology is of "tropical origin" as pointed out by Büdel.

Specific points

Line

1 Title. For me, the important message in this paper that is new and exciting is that the finest clay fractions from exhumed saprolites have yielded K-Ar ages from the Triassic. Inheritance of Mesozoic landscapes is widespread on many cratons so that finding is not novel. Neither is the claim that the strandflat is an exhumed Mesozoic surface (Olesen et al., 2013). The authors may wish to change the title to focus on the ground breaking aspects of the paper.

We appreciate that the reviewers considers the method ground breaking and see the point raised here. However, in this study we combine different methods and K-Ar dating of saprolite and traces of authigenic illite is an important step to constrain this process. K-Ar dating is not the sole focus of the manuscript and we would prefer to keep a more general title highlighting the results: **The inheritance of a Mesozoic landscape in western Scandinavia**. In fact, we focus the whole paper more on the novelty of the methodological approach and implications and less on correlating landscapes, genesis and age.

7 & 12 Check capitals - **Done**

24 Perhaps "mainly thought"? Some saprolite occurrences have been regarded as Pleistocene in age (Sørensen, 1988). - **Done**

25 Weathering in Fennoscandia has a range of ages. - **Changed**

27 Is this first order geomorphology really the strandflat? - **Changed**

30 Do not the stratigraphic constraints suggest a younger age, before the Late Jurassic or Late Cretaceous? – **At the minimum we show that there is Mesozoic weathering, and given the tighter constraints provided by Ahlberg's studies close to Ivö and the Triassic spores in 16/1-15 well core we think the K-Ar data are reasonably well constrained. We leave the sentence as is.**

31 Is it not the age determinations that point to this conclusion? – **we have rephrased this sentence**

37 This paragraph is concerned with gross morphology of the Norwegian passive margin whereas the paper is more directed to the strandflat. – **Agree, both Hall and Egholm points at this. We have moved away from the initial sweeping age determination over several Norwegian landscapes and focus on the method and dating of the strandflat.**

38 Perhaps the word controversial is sufficient? – **the whole paragraph has been rephrased**

38 Is first-order needed here? – **the whole paragraph has been rephrased**

40 Over large areas? Do planation surfaces control the first order geomorphology? – **the whole paragraph has been rephrased**

45 Interfluves are lowered through time and so are dynamic features. It has been argued however that the plateaux are inherited planation surfaces formed before uplift. These planation surfaces are potentially old forms, although slow erosion continues to reshape them. – **the whole paragraph has been rephrased. See point on focus on dating method and strandflat**

48 In response to denudational unloading? – **the whole paragraph has been rephrased**

50 and shaped by glacial and periglacial processes? – **the whole paragraph has been rephrased**

52 dynamic forms of Pleistocene age? – **the whole paragraph has been rephrased**

53 Space could be saved here by deleting this sentence. – the whole paragraph has been rephrased

61 I suspect that neither of these references mentions joint valleys or cnoc and lochan. – Agree, wrong references - fixed

63 This paragraph should perhaps maintain a focus on the strandflat.- agree, the paragraph has been rephrased

71 The authors may wish here or at line 107 to mention any previous attempts to use this K-Ar method to date weathering or alteration clays. – Agree we have added a paragraph on similar attempts outside of Scandinavia

75 Bømlo lies quite close to a Jurassic outlier. - Agree, added to map in Fig. 1

82 The Campanian falls within the Upper Cretaceous. The sediments at this site are early Campanian and so date from closer to 80 Ma. – Agree, changed

84 This paragraph does not inform the reader about the nature of the saprolite at Utsira. – the Utsira drill cores are painstakingly described by Riber et al. (2015), but we added a sentence describing the saprolite from these unique well core samples.

99 Overlie - changed

99 Does the granodiorite have an absolute age determination? – added including references

107 This is an important paragraph. This might be the place to point out the novelty of the application of the technique to weathering clays. – added a sentence on this

116 To assign an age, does not the clay fraction need to be entirely homogeneous and authigenic? – That would be an ideal case. However, with careful analysis of the different grain size fractions we argue the data are robust. See also comments to points raised by the anonymous reviewer.

126 This statement on the weathering origin of illite needs a reference? The alternative diagenetic origin of illite could be mentioned. – added a sentence on polytypes and references

129 This sentence appears to cover two separate points: illite is unstable and only small amounts of fine illite are needed to date authigenesis. – Agree, this is a poorly phrased sentence based on a misunderstanding. We re-read the section in the Meunier Illite book on near surface illite stability. Meunier only mentions that illite is known to be unstable in modern agriculture soils, which necessarily is not applicable to saprolite. We have deleted this sentence.

132-135 This sentence sets out what could be regarded as the most novel and important aspect of the paper and so should appear earlier and in the abstract? – we added a short version of the sentence to the abstract

138 Should the word weathered be here? – corrected

147 In similar rather than at the expense of? - corrected

160 The implication here seems to be that a mixed clay assemblage including authigenic illite is dated. See my comments above on reporting ages from mixed size fractions. Why is the authigenic illite necessarily of weathering origin? – Please see comments to point 3 above

171 Why is a hydrothermal origin excluded? Zonation is common across linear zones of alteration. Alteration zones may also be reactivated. – the outcrop at Bømlo does not appear as hydrothermal alteration. The small scale morphology of the saprolitic outcrop, including rounded joints and corestones (“woolsacks”), closely resembles saprolite for example at Ivö. We added a sentence on field appearance to the text.

178 The age of kaolinisation at this site generally has been regarded as ~100 Myr younger (Gilg et al., 2013). – we added a paragraph discussing possible age correlation with Rhaetian kaolin offshore Ivö (Ahlberg papers).

188 The SI indicates that the Bømlo occurrence is not clay-rich nor highly kaolinitic. It is unclear how clay-rich or kaolinitic (or thick) is the Utsira weathering profile? So whilst these occurrences may be the roots of Triassic kaolinitic weathering profiles, this needs to be established. – The Utsira High well cores are described in detail by Riber et al 2015 so we omitted that to save space. What we try to say

here is that we evidence for a Triassic regional saprolitization event in SW Scandinavia, that might have played a role in shaping some of the landscape we see today. In addition we see, by using quartz grain fingerprinting (SEM-CL), that the quartz grains in the Triassic saprolite is identical to the Draupne fm. quartz grains indicating recycling of saprolite into the sediment system.

190 The word abrasion might be omitted. - corrected

205 As argued earlier, similarity of form provides only weak support for similarity in age. – That is true. However, similar morphology and similar apparent (K-Ar) age makes it tempting to speculate that the onshore and offshore landscapes have a common origin. Low temp. geochronology of Ksienzyk et al. (2013) shows that both areas probably were subaerially exposed in the Triassic.

206 The concluding paragraph makes a huge leap from a set of dates based on a novel application of the K-Ar dating technique that indicate a late Triassic age for authigenic illite at one site on the strandflat to a bold statement on the age and origin of the entire strandflat and on the post-Caledonian tectonic history of western Norway. – Agree, this was an unjustified leap as also concisely pointed out by David Egholm. We have rewritten the Introduction and Discussion, focusing on the K-Ar technique and the age of the strandflat. We have omitted inferences on gross morphotectonic history of Scandinavia, other than that the strandflat probably only have experienced limited tectonic movement since the Mesozoic, which is also supported by recent low temperature geochronological data.

References

- Bøe, R., Fossen, H., Smelror, M., 2010. Mesozoic sediments and structures onshore Norway and in the coastal zone. Norges Geologiske Undersøkelse Bulletin, 450, 15-32.
- Burley, S., Flisch, M., 1989. K-Ar geochronology and the timing of detrital I/S clay illitization and authigenic illite precipitation in the Piper and Tartan Fields, Outer Moray Firth, UK North Sea. Clay Minerals, 24, 285-315.
- Gilg, H.A., Hall, A.M., Ebert, K., Fallick, A.E., 2013. Cool kaolins in Finland. Palaeogeography, Palaeoclimatology, Palaeoecology, 392, 454-462.
- Godard, A., Simon-Coinçon, R., Lagasquie, J.-J., 2001. Planation surfaces in basement terrains. In: A. Godard, J.-J. Lagasquie, Y. Lageat (Eds.), Basement Regions. Springer, pp. 9-34.
- Hamilton, P., Kelley, S., Fallick, A.E., 1989. K-Ar dating of illite in hydrocarbon reservoirs. Clay Minerals, 24, 215-231.
- Olesen, O., Kierulf, H.P., Brønner, M., Dalsegg, E., Fredin, O., Solbakk, T., 2013. Deep weathering, neotectonics and strandflat formation in Nordland, northern Norway. Norwegian J. Geol, 93, 189-213.
- Sørensen, R., 1988. In-situ rock weathering in Vestfold, southeastern Norway. Geografiska Annaler. Series A. Physical Geography, 299-308.
- Thomas, M.F., 1989a. The role of etch processes in landform development. I. Etching concepts and their applications. Zeitschrift für geomorphologie. NF, 33, 129-142.
- Thomas, M.F., 1989b. The role of etch processes in landform development. II. Etching and the formation of relief. Zeitschrift für geomorphologie. NF, 33, 257-274.

Adrian Hall - Thank you for a very constructive review!

Reviewer #2 (Remarks to the Author):

Review of "The inheritance of a Mesozoic landscape in western Scandinavia" by Fredin et al.

This study presents new age constraints on saprolites along the coastal areas (the strandflats) of the

western Scandinavian mountains. The results suggest that the saprolites studied are exceptionally old, dating back to the late Triassic period (~200 Myr ago). The finding hence indicates long-term stability of the strandflat areas in spite of significant erosion further inland and deposition of sediments offshore during several hundred million years.

Some general comments:

The results presented here are important, and I think the study represents a significant step forward for our understanding of long-term landscape evolution in Scandinavia. It also advances a novel quantitative method based on K-Ar dating. Furthermore, the text is generally well written and I enjoyed reading it.

For these reasons I believe the study will attract significant interest. In my opinion the results presented here deserve to be published in Nature Communications.

Yet, there is an urgent need to revise the introduction and the discussion of the results, as it is hampered by misunderstandings of previous work (see below). Once these misunderstandings are corrected, however, I think that the authors can significantly improve the discussion and strengthen their conclusions, but it requires that the authors critically rethink the consequences of their findings. I hope the authors will find my comments useful in this process.

The authors generally discuss their findings against two existing hypotheses for development of the Scandinavian topography. One (the 'peneplain-hypothesis', refs 2, 5, 11-13) assumes that the current topography was formed by a series of late-Cenozoic tectonic events that lifted surfaces from sea level (fluvial base level) to their present elevation (up to more than 2 km). It states that the current topography in Scandinavia is generally young (late-Cenozoic), but may include older elements that have been buried and later exhumed by late-Cenozoic erosion.

The other hypothesis (the 'Isostasy-Climate-Erosion (ICE) hypothesis' ref 6, of which I am a coauthor) is simpler and suggests that the topography is a long-lived slowly eroding and partly collapsed remnant of the Caledonian Mountain belt, which first formed more than 400 Myr ago. This hypothesis does not call for recent tectonic uplift events but relies on climate sensitivity of surface processes to explain variable sediment flux to the surrounding basins.

The misunderstanding I mentioned above relates to the ICE-hypothesis:

In lines 51-52 it reads: "The main implication of this model is that all landscape elements essentially are of the same Pleistocene age"

This statement is simply wrong and very misleading! It is very important to correct it because it unfortunately complicates the discussion and sends the readers in a wrong direction.

We agree with the above critical comments from David Egholm, and Adrian Hall is touching upon the same issue. The *Introduction* and *Discussion* came out as very polarized in the initial manuscript submission. After testing the method at two test sites, we attempt to date one unconstrained saprolite and hence landscape, namely the strandflat. The connection we make with the recent and very vigorous discussion ('peneplain hypothesis' vs. 'ICE-hypothesis') is that we provide numerical evidence of an ancient landscape close to sea level in western Norway (see reference Twidale (1998) - "Antiquity of landforms"). It is clear to us now, as both Egholm and Hall, point out that it is not justified to extrapolate these findings to regional morphotectonic history. We have rewritten the Abstract, much of the Introduction and Discussion/Conclusions accordingly. As a result, we now focus

strictly on the novel use of the method to date saprolite and landscape, and implications for the strandflat landscape.

The ICE-hypothesis does indeed suggest that the high plateaus in Scandinavia were affected by glacial/periglacial erosion (which does not seem completely farfetched when considering that many of them likely have a >10 Myr glacial/periglacial history). Importantly, however, this does not mean as stated in line 51-52 that all landscape elements in Norway were formed in the Pleistocene, not even the high plateaus (ref 6 suggest they started forming in the Oligocene) and certainly not the strandflats!

In contrast, the ICE-hypothesis proposes that the first-order topographic structure of western Scandinavia is a remnant of the Caledonian mountain belt, which over time has been gradually destroyed by rifting and long-term climate-sensitive erosion. According to the ICE-hypothesis, the coastal areas in particular, have been more or less stable for hundreds of millions years (since rifting), because vertical displacements were dominated by isostatic adjustments causing rock uplift in-land and subsidence off-shore, leaving the coast zones as a relatively stable tipping line (influenced however by variable sea levels and sediments that may have piled up in near the coast). There is a section in ref 6 dedicated to the long-term stability of the coastal areas. I suggest the authors reread this before revising their manuscript.

In short, I think the old saprolite ages reported here are very much in agreement with the ICE-hypothesis.

Corrected, we have significantly modified the manuscript in this section. We now strictly focus on the strandflat and its potential off-shore correlatives. We agree with the reviewer that our data is consistent with relative tectonic stability of the strandflat (supported also by low temp. geochronology of Ksienzyk et al. 2014, and two theses by Utami and Wosnitza). Obviously the Utsira High has subsided more than 2 km since the Mesozoic, so from that perspective the coastal area, and strandflat, can be considered a hinge zone as outlined by the reviewer and ref. 6 (Nielsen et al., 2009).

In contrast, it is much more difficult to reconcile Triassic saprolites with the other hypothesis including Late-Cenozoic uplift of peneplains (refs 2, 5, 11-13). According to this hypothesis the uplifted peneplains must become older with elevation, such that the highest represents the oldest preserved landscape and the first uplift event (see age distribution of surfaces in ref. 12).

I believe that the present study in combination with previous work (Goddfellow 2012, Goodfellow et al. 2014, not cited in the manuscript) on younger saprolites at higher elevation demonstrates that it is the other way around: the oldest (most slowly developing) landscapes are near the coast and the youngest are higher up where erosion processes are more active (which, by the way, the ICE-hypothesis would also predict).

The authors fail to discuss this properly because the misunderstanding about Pleistocene landscapes and the ICE-hypothesis stands in the way. In my opinion, the findings reported here first-of-all demonstrate that post-orogenic mountain belts are long-lived features dominated by overall slow erosion processes and weathering (except for where ice-sheets choose to erode) and this finding is important to publish.

Corrected, we have thoroughly rewritten the manuscript so that it is better aligned with this line of reasoning. We indeed agree that the highest likelihood for a landscape to survive hundreds of millions of years is if it is situated close to the sea level, and it certainly also helps if it is covered by protective sedimentary rocks through long periods. This seems to be the case for the Bømlø strandflat area.

Some more specific comments:

line 27-28: How can a handful of samples from the strandflat area and offshore (that is at and below sea level) deliver "... evidence that first order geomorphology of western Scandinavia is a relic from Mesozoic times"? By first-order geomorphology of western Scandinavia I understand the overall shape of the mountain range, but surely this study is not dating this.

This comment actually points to a general problem in this study: dating weathering is not the same as dating a landscape. Weathering landscapes in particular are known to evolve by slow surface lowering under steady-state geomorphic conditions (see for example Anderson 2002, *Geomorphology* 46, p. 35-38). In this case, one can time the weathering but it does not make sense to define an age of the landscape. Assigning ages to bedrock landscapes has been a popular activity in Scandinavia, but I think that it should be done with more caution.

- Perhaps "first-order geomorphology" is incorrectly used in the manuscript. However, the strandflat is certainly a landscape type of great importance and as we try to show potentially of great age.

line 29: following on from the last comment, it is not clear how basement exposure and weathering are correlated in these saprolites. Can the authors provide an estimated upper limit to the maximum saprolite thickness, and explain how exposure links up with chemical weathering?

line 32: How do we know that the strandflat landscape "initiated" in the Triassic? This study's results demonstrate that the strandflat experienced weathering then, but how do we know that it had just started?

-Based on arguments outlined above in response to points raised by Adrian Hall we believe that had it started earlier (significantly earlier), then the ages would be older than Triassic. Illite, especially the tiny crystallites, is believed to track a specific geological event. The Triassic ages record a steady state evolution, where a balance was found between comminution of the old protolithic component and the authigenic growth of the synweathering part. So, I would exclude that the process started much earlier than what we date.

line 48: consider writing "...slow isostatic crustal uplift..." - the whole section is rewritten.

line 51: The authors must correct this misunderstanding and consider its implications. - Corrected, the manuscript is thoroughly revised.

line 63: I do not agree that this study is dating the strandflat landscape. The authors are dating the formation of clay minerals found in the landscape (see point above for lines 27-28), which is not the same as the landscape (see comment above). - We elaborate more on this in the manuscript and to points raised by Adrian Hall above. We believe that K-Ar dating of saprolitic illite is a good proxy (the best we currently know of) for weathering/etching age.

line 72: I think this test is a clever approach and the method description is a real strength of the study. But I wonder if the method would also work in younger saprolites that have not yet developed large amounts of secondary clay minerals? - An interesting question that we unfortunately cannot explore in this manuscript. We are currently experimenting and attempting to date grussy saprolites (of hypothesized younger age). It is often a problem to obtain enough clay (and hence 1M polytype Illite) in the finest fractions to be able to measure Ar and K reliably. This will hopefully be the topic of a future publication.

line 98: Can the authors say more about the overlying strata that was stripped in the late Cenozoic? Does the saprolite give any constraint on their thickness (from e.g. compaction)?

- Surprisingly little can be said on the overlying strata based on saprolite characteristics. The best estimate on sedimentary overburden comes from Ksienzyk's, Utami's and Wosnitzer's low temperature geochronology work, where they estimate Cretaceous reheating to maximum 50°C in coastal Norway. This corresponds to about a km of burial. Apparently this is not enough to trigger diagenesis and/or compaction. It is interesting to note that saprolites both at Ivö and at Rønne (Bornholm) where the overlying strata still is present, also show no evidence of diagenesis and compaction. As a sidenote, we visited Rønne in the spring 2016 for sampling but the results are not yet available. There is also one site on Bømlo where there is overcompacted, sub-glacial till, slightly tectonizing the underlying saprolite. It is somewhat of a paradox that the subglacial till apparently is compacted and more competent than the saprolite. We have also unpublished observations from saprolites in well cores outside of Vesterålen/Lofoten that appear overcompacted.

We expand on this in the manuscript using the published low-temperature geochronological studies

In order to make any sense (see comment above on age-distribution in ref 12), the peneplain model often has to form a landscape, bury it, and exhume it again without the erosion processes changing the form of the original buried landscape. It is difficult to understand how that works, but of course a strong contrast in erodability between the cover and the buried landscape would help. It would be good to see this discussed more here.

- If we call the strandflat an exhumed planation surface then the contrasting erodability between overlying strata /saprolite and bedrock is indeed the best explanation of how the gross weathering morphology has been preserved. However, as we point out also in response to a similar concern raised by Adrian Hall, we also acknowledge that Pleistocene glaciations (and perhaps other surface processes - marine, fluvial,...) potentially not only have removed the overburden but also polished and accentuated the strandflat landscape. In other words; the strandflat is not an entirely pristine, untouched Mesozoic landscape. The gross morphology is akin to an etch surface, but with a Pleistocene polish. We elaborate on this in the revised manuscript.

The results section is convincing and very well written. - Appreciated

line 161: I wonder from what criteria the sampling sites were chosen? Is it possible that the Triassic age plateau emerges because only very mature saprolites were included in the study? Many weathering profiles in Scandinavia seem to be less mature (again, see Goodfellow's work). Would they challenge the image of a Triassic age plateau if they were measured? Overall, I think that the study needs to include more info on previous work on weathering profiles in Scandinavia, and a broader discussion on how representative the samples measured here are.

- As explicitly said in the text, saprolite occurrences in Scandinavia are generally few. Our field work was initially aimed at simply "finding" saprolitic sections in SW Norway. It is honestly unrealistic to think to select saprolitic dating targets out of an already limited number of accessible outcrops based on a very comprehensive list of textural, geomorphological or mineralogical criteria. In this study, like in the many more studies that we are currently conducting, we were often forced to sample and try to date whatever saprolite or saprock we would find at a given locality. This, by the

way, also explains the fact that some samples did not yield the finest < 0.1 micron fraction, indeed because there was only a limited amount of sample to collect.

In detail, however, the Bømlo site was also selected because it appeared to us as an onshore equivalent to the quite well explored Utsira High basement/saprolite/sediment system (detailed by Riber et al. 2015) sampled from drill cores. We wanted to test if the Bømlo saprolite would also be of sub-Jurassic age after that we had validated the method on Ivö and Utsira High. There are several sites with so-called 'grussy saprolite' in Scandinavia, which commonly is correlated to the Plio-Pleistocene (northern Finland - Gilg et al., 2013; northern Norway - Peulvast, 1985; southern Norway - Sørensen, 1988; southcentral Sweden and northern Europe - several references by Migón & Lidmar-Bergström). Indeed Goodfellow et al., (2016) show that biotite oxidation and thorough collapse of a rock column can happen very rapidly producing coarse, grussy saprolite.

In this study we thus tried to focus on the clay rich, likely older saprolite type. We have manuscripts in the workings where we try to characterize different saprolites in different regions, but this work is outside the scope of this study. However, we add some references and reflections on this in the Introduction

It should perhaps be noted that we have not attempted to date the blockfields of Scandinavia ('paleic surface'), as we generally agree with the reviewer and his research group that these are dynamically evolving landscapes and if there ever was a palaeosol on the blockfields it is likely gone today due to erosion. It would probably be difficult indeed to find dateable material on the blockfields (as also shown in several papers by Goodfellow).

line 200: The "cnoc-and-lochan" landscapes likely form because of regional differences in bedrock fracture density. This difference may influence the pace of weathering and fluvial/glacial erosion in the same way, so it is not unlikely that different surface processes can form the same type of landscape, at least on length scales that can be resolved by seismics.

- We agree in principle, and respond to this according to the same point raised by Adrian Hall. It is true that a similar morphology does not imply the same origin and age. As Adrian Hall points out, geomorphology is often polygenetic and diachronous. However, apparent similar morphology, and K-Ar dated to a similar time allows us to speculate that there was a widespread Triassic etch surface in large areas of SW Scandinavia (large parts now probably offshore).

line 212-214: I strongly disagree with this conclusion, and I cannot see how the authors can arrive at this given the sentence before. At least they should explain it better.

- we have significantly revised this section and the manuscript as a whole, and do not any longer conjecture on broader morphotectonic relationships.

Supplementary info: The additional figures (in particular the photos) are great and very useful. I think that the format of Nature Communications allows for including some of them in the main text.

- Thank you, we will consider doing this.

I hope that the authors are given the opportunity to revise the manuscript, as they present important new results. The results do, however, deserve to be discussed in a more convincing way. The authors are welcome to contact me, should they have any questions. - Appreciated

David Egholm

- Thank you for a very stimulating review!

Reviewer #3 (Remarks to the Author):

A. Summary of the key results:

An interesting paper. K-Ar geochronologic methods are used to establish the age of pedogenically (authigenic) illite in saprolite/weathered rock during Mesozoic era. These dates are used to argue for a long term development of the standflat landscape in western Scandinavia. They observed a decrease of measured K-Ar age in grain size in two saprolites stratigraphically constrained (Ivö and Utsira) and in one saprolite (Bømlo) not constrained to develop their model for stand flat formation. At first glance, these data make sense and lead to the paper being considered by Nature. The paper is fairly well written and provides a good context for wide multidisciplinary readership. There is one stratigraphic miscue (e.g. line 82, "middle" Cretaceous - See Gradstein et al cited in the SI). I am more focused on the K-Ar systematics than landscape formation.

- Corrected stratigraphic mistake. Also pointed out by reviewer Adrian Hall

B. Originality and interest: if not novel, please give references:

The use of K-Ar geochronological data to study landscape development is novel and original. Vasconcelos' work is the only work coming close to this effort. The development of terrestrial climate proxies is one of more difficult aspects of the study of ancient climates. This paper opens another possible tool for developing climate proxies. This paper will potentially attract a wide readership.

- Thank you, in the manuscript we indeed cite Vasconcelos extensively where we explore what has been done before in the field. As we point out in the manuscript, other studies have often focused on younger supergene minerals in other climatic and geological settings (outside of Pleistocene glaciations).

C. Data & methodology: validity of approach, quality of data, quality of presentation:

The extraction of authigenic (diagenetic, or even pedogenic) illite from rocks, sediments, and soils having mixed provenance is very difficult work. I can appreciate the painstaking efforts in the areas of STEM and characterization to show the presence of this fine-grained illite (and smectite, if it is illite-smectite, see following point) in the presence of other phases including K-bearing phases in some instances (Utsira). I bring up some points needing some clarification.

For the Bømlo saprolites, it appears the K-bearing phases giving rise to the K-Ar ages are smectite (and albite/anorthite?) in the fine fractions (mica is seen only in the coarse fraction). The finest fraction of this Bømlo2 saprolite gives the lowest age and interpreted as authigenic. Could this smectite be an illite-smectite per XRD? If yes, then how does the presence of I-S condition the interpretation and conclusions for the formation of these phyllosilicate minerals? Still authigenic? Pedogenic? Transformation?

The reviewer raises an important issue with the K sources in the separated clay fractions in the Bømlo saprolites. The traces of Albite (Na rich) and Anorthite (Ca rich) feldspars are excluded as meaningful potential source of K because of their composition and therefore we do not consider the radiogenic Ar to be derived from albite or anorthite. As for the finest fraction of Bomlo 2, as stated in the text, it was SEM analysis which disclosed the presence of tiny amounts of authigenic illite, given that that amount is beyond detection at the XRD.

The interpretation of the sampled section (Bomlo 2, 3 and 4) is that of a progressive textural evolution of the saprolite from less evolved end-members at the edge of the outcrop towards more mature saprolites in the core of the section. This is demonstrated by the kaolin amount increasing significantly

away from the fresh host rock as the amount of smectite decreases. This trend is most simply read as being consistent with progressive alteration during saprolitization of the host granodiorite, where illite formed at the expense of K-feldspar, plagioclase and biotite in association with smectite for low weathering degrees and kaolinite for more extreme alteration. Therefore, we read the possible presence of mixed layers IS as part of the primary processes that caused alteration of the granodiorite. It is, in our opinion, a continuum. And along this continuum we expect to find and, indeed do find, different and yet coexisting clay types.

The Ivö saprolite shows kaolinite and small amounts of illite-mica. Is this illite-mica a 1M polytype mica while the other mica is 2M1 as per Table S11?

What we know of the polytype composition of the analysed samples is clearly reported in the table. We can only interpret the data. Due to small sample size and also sample characteristics we were not able to categorize polytypes from the Ivö saprolite samples. As outlined by Zwingmann et al. (2010) and discussed by Haines and van der Pluijm (2008), 1M and 1Md are not really distinct polytypes but rather end members of a spectrum, and it is more appropriate to consider them as a single 1M/1Md polytype. Based on X-ray diffraction (XRD) analysis illite (2M1) cannot be distinguished from muscovite (also a 2M1 polytype). We therefore labeled this mineralogy as illite/mica in table S1. We interpret these illite to be of 1M and authigenic origin as supported by petrographical investigations.

The K-Ar ages of the Ivö saprolite are most supportive of their argument. Even so, you have potentially two K-bearing clays (smectite, some K is fixed or is it illite-smectite with more fixed K) and illite/mica (1Md illite?). Are both the smectite and illite-mica authigenic?

Yes, that is our interpretation. It is not crucial in these samples (with this very mineralogy) to know where the K is hosted. As long as one can exclude very significant contributions of protholitic phases, the age of the finest can be taken as the age of authigenesis. And the “age vs. grain size” inclined spectra offer, in combination with the details derived from sample characterization, a good way to tell apart the input of authigenesis from the input of detrital/protolith phases.

For the petrographic work, I follow the argument for authigenic illite based on the data presented including the K-Ar data. It is possible to see mica or weathered mica in kaolin-rich saprolites as evident in the rocks nearby my home.

The reviewer raises an important issue if weathered mica is potential included in the separated mineral fractions. The radiogenic isotope systematics of sedimentary rocks and clay stones are complex due to the intimate mixture of minerals of different origins such as detrital phases, potentially from a variety of sources, as well as authigenic minerals (see summary Clauer and Chaudhuri, 1995; and references within). Consequently, it is often difficult to unambiguously interpret the measured ages. One determining aspect of dating sedimentary rocks is the initial primary sample preparation method. Liewig et al. (1987) compared two different sample disintegration methods (grinding versus gentle heating freezing method) on clay minerals and its influence on the obtained isotopic data. Special sample preparation techniques involving freeze-thaw disaggregation to avoid overcrushing and extensive size separation to reduce the amount of detrital phases can reduce sample disintegration methods artefacts (Clauer et al., 1992). Progressive size reduction down to submicron size fractions (<0.1 µm or finer) increases the proportion of authigenic clay phases in the clay component and minimizes contamination and suggest that the most reliable isotopic ages for authigenic clay minerals are obtained for the finest size fractions. This has also been documented extensively in the case of

authigenic synkinematic illite from brittle fault rocks (e.g. Viola et al., Nat. Comm., 2016; Torgersen et al., Terra Nova, 2015).

As we used in this study a cryostat sample disintegration method within this study we consider contamination with weathered normally larger micas, which can be eliminated with the extensive size separation, as minimal, although it cannot be ruled out fully. In addition, no larger K-rich particles could be observed during the TEM studies, which supports a clean separation process.

The data that would have cinched this argument would have been to show illite being weathered from mica or K-feldspar in these saprolites (e.g. Elliott et al, 1997, CHEM GEOL - Elsevier just posted a cleaner version of this paper). Illite polytype analyses would have helped. It would have been useful to post the XRD patterns in the SI data.

We did our best with the available sample material and think it is not necessary to supply 16 XRD spectra and numerous SEM and TEM images. We normally include some representative images/spectra as is done in the manuscript. We could discuss if it is feasible to upload that kind of data to a digital repository at Nature Communications if the paper gets published.

In other words, I am questioner of the pedogenic/authigenic origin of this illite. The K-Ar ages yield younger age signifying either loss of Ar by weathering a muscovite mica from bedrock or creation the authigenic formation of illite. Can a transformed or weathered mica from bedrock be ruled out? If micas are being weathered to illite, is this weathering congruent in terms of K and Ar behavior in this illite/mica? Is more parent or more daughter lost/gained? I would gather more daughter (Ar rad) is lost in weathering.

We believe that we are very consistent in our line of thought and have already explained why we think that the finest fractions are mostly authigenic whereas the protolithic/detrital component increases in the coarser fractions. Also, we have provided many specific references to recent papers that deal with this subject, even though within the system of brittle fault rocks.

In addition to the comments already made above, however, we can further rebut to this point by confirming that Ar loss in clays is often debated as a potential reason for the younging age trend with the finer fractions. This issue has already been in part dealt with by some of us in the past and our take is available in published peer-reviewed papers (e.g. Zwingmann et al., 2010 and Torgersen et al., 2014). In summary, for the reasons discussed here below, we do not think that radiogenic ⁴⁰Ar volume diffusion can affect significantly the age-grain size spectra.

Dating assumes that no isotopic re-equilibration has occurred since the dated minerals formed. However, Ar loss could be due by exposure to temperatures at or above the formation temperature for considerable time intervals, thus leading to a partial or full reset of the system, with mixed ages being the end result.

The effects of thermally-induced volume diffusion can be best explored by a numerical modelling approach. Unfortunately, however, Ar diffusion modelling in clays is not common, due to conceptual difficulties arising from their fine grain size and poorly constrained diffusion parameters. Thus, a simple transfer of modeling results obtained for relatively coarse mineral grains (such as biotite or muscovite > 100 μm) during long-lasting and gradual cooling following regional metamorphism, is certainly not ideal to clays formed within low-T systems such as saprolites dominated by important fluid-rock interaction.

Potential partial Ar resetting within clay-size crystallites has been previously done with an Ar diffusion code by, for example, Zwingmann et al. (2010), based on the diffusion model by Huon et al. (1993), and Torgersen et al. (2014), based instead on the script DIFFARG by Wheeler (1996) implemented with the most recent diffusion parameters by Harrison et al. (2009).

Temperatures of up to 150 °C have been found to be negligible for diffusion within a cylinder and plate shape particle with 0.1 µm grain size. Instead, up to 42 % of radiogenic Ar could be lost by diffusion if the overprinting temperature reached 200° C for a 0.1 µm cylindrical clay particle and up to 22 % for a plate shape particle with the same grain size within a 1 Ma overprint time. These high temperatures are unrealistic in the case of the studied saprolites, as shown by the available low-T geochronological results for the studied areas.

Therefore, and as no systematic shift or disturbance was observed in the presented age data, we consider Ar loss by diffusion an unlikely process in the case of the dated saprolites.

On a different front, we also realize that dating of authigenic illite formation is an imperfect proxy for the actual saprolitization age, simply because weathering is a time transgressive process. This is an issue also raised by the other two reviewers. However, we believe that true authigenesis in the saprolites reflects a steady-state process, wherein in-situ clay formation keeps pace with the continuous degradation of the fresh host rock and possibly exhumation/erosion of the saprolite.

D. Appropriate use of statistics and treatment of uncertainties:

The treatment of uncertainties is good for the K-Ar data. The measurement of high percentages of radiogenic argon is impressive for the low K samples from the Bømlø2, Ivö samples. The recording of mole Ar-40 rad/g out to five significant figures tells me a microbalance or high-quality analytical balance was used. No measurement of interlaboratory standard noted for K-Ar data. Not enough information is provided about the semi-quantitative XRD techniques.

- Done, the methods section has been updated and we now include K-Ar interlaboratory standard data and airshot data. The XRD methods have been included as well.

E. Conclusions: robustness, validity, reliability

The key to this paper is accepting the interpretation of the petrographic data as well as some more detailed exposition of the illite polytypes in the saprolite matrix materials. The authors could have stressed better the alignment of the measured ages and the stratigraphic constraints.

We agree, in the revised manuscript we discuss more in depth how the measured ages compare to stratigraphic constraints.

F. Suggested improvements: experiments, data for possible revision:

I agree with the authors that laser- Ar-Ar might not be useful here. Their approach is the best that anyone can do with the available techniques. Having the XRD data available answers questions about the smectite (is it I-S?). Examination of the feldspars for its weathering to illite and ruling out bedrock sources of illite/mica help this argument tremendously.

We have added SEM imagery from Ivö that aid in this interpretation.

G. References: appropriate credit to previous work? OK.

H. Clarity and context: lucidity of abstract/summary, appropriateness of abstract, introduction and

conclusions
Comprehensible paper.

- Thank you for a constructive and thorough review!

Reviewers' comments:

Reviewer #1 (Remarks to the Author):

The manuscript has been substantially improved in its latest version. The significance of the paper is now clearer and there is little doubt now that the findings will be of wide interest.

My main concern remains with the conclusion that the strandflat is largely an exhumed Mesozoic feature. I think this is likely to be wrong, as I indicated in my original review. Yes, the Jurassic remnants indicate Pleistocene exhumation of Mesozoic bedrock surfaces along parts of the coastline. Yes, the new age determinations support a major phase of weathering in the Triassic. A dated saprolite of that age is present at one site on the strandflat at Bomlo. So at the current erosional level we have fragments of sub-Mesozoic weathered bedrock surfaces. The strandflat however is cut across these surfaces and so is a younger, likely Pleistocene surface, formed close to sea level along hundreds of km of the Norwegian coast.

The arguments and language should be checked again by the authors as there is still a tendency towards over-extension of the evidence and also some loose expression in places.

Adrian Hall

Reviewer #2 (Remarks to the Author):

Comments to revised version of "The inheritance of a Mesozoic landscape in western Scandinavia" by Fredin et al.

Overall, I think the authors have done a fine job in revising this manuscript. The updated version is much improved, and I feel that most of my initial comments have been addressed in a satisfying way. Importantly, the misunderstanding of previous work that I pointed to in my initial comments is now out of the way.

I have only few comments left, which I hope the authors find useful:

I think the authors could still improve the text to better clarify how the saprolite-age relates to landscape-age, or alternatively soften statements a bit. The arguments regarding this remain a bit vague, but considering the manuscript title and the concluding statements, linking saprolite-age to landscape formation seems really important.

The study convincingly demonstrates that the saprolites found at few locations along the strandflat are of Triassic age. The authors argue that the gross morphology of the strandflat was also formed then, and that it was subsequently buried, and finally exhumed by Neogene/Pleistocene erosion without changing the original strandflat landscape.

On one hand, one can argue that the saprolite finding supports this theory, at least it demonstrates that subsequent erosion has not fully erased the saprolites everywhere. On the other hand, one can ask: If Triassic weathering really formed the current strandflat landscape, why are saprolites not more frequently found in it?

Overall, I'm still not convinced that the gross morphology of the strandflat, as it looks now, dates back to the Triassic. It might, but I do not think that a small number of Triassic saprolites fully demonstrates this. It is true, that the strandflat looks like a weathering surface (line 298), but it also looks like a zone of areal scour (line 69) so it is risky to rely on morphology.

However, the main point regarding long-term stability of the coastal area (dismissing km-scale erosion or deposition) stands clearer now (in lines 286-297), and to me this is the most convincing and important finding. In fact, I would recommend that this point be reflected in the title somehow instead of the "Mesozoic landscape".

Another point regarding this subject is that the last sentence (line 310) is rather imprecise. What

exactly is an "ancient landscape"? To some it might be a landscape just pre-dating the latest glaciations (as often demonstrated using cosmogenic nuclides), to others it means something older, or a surface that has not eroded for a long time. Please be more specific.

This leads to a final (minor) comment: the preservation of landscapes in glaciated terrain is often associated with cold-based/non-erosive part of an ice sheet, but the ice-sheets were presumably not cold-based along the coasts of Scandinavia. It would be good to see this discussed a bit – i.e. in few sentences that describes known conditions for landscape preservation under ice sheets.

Otherwise, I think that the text reads really well, and I look forward to seeing it published.

David L Egholm

Reviewer #3 (Remarks to the Author):

I read the revised draft, the reviews, and the responses to the reviews. My review and comments focus on the K-Ar technique and the clay mineralogy of these saprolite samples.

The revised text is an improved version all through the text. The authors provided good answers to all questions.

The point sticking me is the presence of smectite in the finest fractions of the Bromlo2 while Ustira and Ivo have some illite or mica in the finest fractions. That smectite must have some fixed K and evidently enough for a useful mass spectrometric analysis (so, yes, you have a more sensitive MS relative to the earlier MS used by many investigators, no need to run an unspiked sample!). The origin of that smectite is a weathering origin? I know the authors go back and say the K-Ar tells the story. The relative similarity of these ages argue for a similar process. Ok, I see it. I advise stressing the fixed K in the smectite and thus a datable phase - given the stress the authors place on the K-Ar ages of these finest fractions. The K value is 0.15 (about what I would guess for smectite). My guess is that a closer look at the XRD pattern for that sample will show mixed layer illite with a small number of illite layers (I am thinking to look for non-integral lower order peak positions or asymmetry of the 17 angstrom peak).

Good analyses on the LP6.

Line 428: need a "b)" somewhere. Lines 444-445 there are two "e)".

All told, this paper will be a very useful contribution. It will show another use for K-Ar analyses. This study is informative with some very interesting (and rare) field context outcrops. The linkage to Triassic pCO₂ is useful and makes the paper very interesting (terrestrial paleoclimates are difficult to interpret). This study builds knowledge both in geochronology, terrestrial climate proxies, and paleoclimatology.

Rebuttal letter (second revision) to "The inheritance of a Mesozoic landscape in western Scandinavia", submitted to Nature Communications.

Author comments are all typed in red font, while reviewer comments remain in black font.

Again we thank the reviewers for encouraging and thoughtful reviews! If we try to summarize the remaining criticism the reviewers seem to think that there still are some doubts as to whether the strandflat landscape at Bømlo really is of Mesozoic origin, or instead if Neogene/Pleistocene erosion is the main landscape-forming agent. We like to think that the current landscape witnesses both Mesozoic and Neogene/Pleistocene weathering, where the primary Mesozoic etching facilitated extensive stripping and erosion in the Neogene/Pleistocene. We have further emphasized this view in the revised manuscript and generally tightened the language, while, at the same time we tried not to over-interpret our evidence.

Reviewer #1 (Remarks to the Author):

The manuscript has been substantially improved in its latest version. The significance of the paper is now clearer and there is little doubt now that the findings will be of wide interest.

Thank you.

My main concern remains with the conclusion that the strandflat is largely an exhumed Mesozoic feature. I think this is likely to be wrong, as I indicated in my original review. Yes, the Jurassic remnants indicate Pleistocene exhumation of Mesozoic bedrock surfaces along parts of the coastline. Yes, the new age determinations support a major phase of weathering in the Triassic. A dated saprolite of that age is present at one site on the strandflat at Bomlo. So at the current erosional level we have fragments of sub-Mesozoic weathered bedrock surfaces. The strandflat however is cut across these surfaces and so is a younger, likely Pleistocene surface, formed close to sea level along hundreds of km of the Norwegian coast.

We argue that the strandflat, at least in the Bømlo area, probably has a Mesozoic origin, in a time when deep weathering genetically also associated with discrete brittle faulting created a saprolite mantle and a bedrock etch surface dissected by joints and faults. This landscape was later readily stripped by Pleistocene glacial and marine processes, although a few pockets of the Mesozoic saprolites escaped stripping basically because they now are expressed as very "deep keels" of the weathering horizon. We think it is important to consider that the Bømlo strandflat has formed in (at least) two different geological episodes, through processes related to deep weathering in a warm climate in the Mesozoic and glacial-/cold climate in the Pleistocene.

By looking at topographic profiles, such as in Figure 2, it appears indeed as a possibility that the strandflat cuts older "envelope surfaces" and thus is younger than the Mesozoic. However, the correlation of topographic profiles is often also considered ambiguous.

We agree that strictly speaking our evidence is only valid for the strandflat at Bømlo, where we have sampled. As a consequence we have changed our possibly overly bold statements that all strandflat landscapes of Norway have a Mesozoic origin. We now say that the strandflat landscape at Bømlo were initiated in the Mesozoic, with a much later Pli-/Pleistocene finish.

This notwithstanding, we maintain that there is the need for the same fundamental geomorphological process to account for similar Mesozoic ages in all of our investigated sites.

The arguments and language should be checked again by the authors as there is still a tendency towards over-extension of the evidence and also some loose expression in places.

We have improved the style and the language and also toned down overly bold statements. We now strictly refer to the "strandflat at Bømlo" and refrain from arm-waving as to the age and origin of other strandflat areas along the Norwegian coast.

Adrian Hall

Reviewer #2 (Remarks to the Author):

Comments to revised version of "The inheritance of a Mesozoic landscape in western Scandinavia" by Fredin et al.

Overall, I think the authors have done a fine job in revising this manuscript. The updated version is much improved, and I feel that most of my initial comments have been addressed in a satisfying way. Importantly, the misunderstanding of previous work that I pointed to in my initial comments is now out of the way.

I have only few comments left, which I hope the authors find useful:

I think the authors could still improve the text to better clarify how the saprolite-age relates to landscape-age, or alternatively soften statements a bit. The arguments regarding this remain a bit vague, but considering the manuscript title and the concluding statements, linking saprolite-age to landscape formation seems really important.

This is also pointed out by Hall, we have now revised the language so that it reads more stringently.

The study convincingly demonstrates that the saprolites found at few locations along the strandflat are of Triassic age. The authors argue that the gross morphology of the strandflat was also formed then, and that it was subsequently buried, and finally exhumed by Neogene/Pleistocene erosion without changing the original strandflat landscape.

On one hand, one can argue that the saprolite finding supports this theory, at least it demonstrates that subsequent erosion has not fully erased the saprolites everywhere. On the other hand, one can ask: If Triassic weathering really formed the current strandflat landscape, why are saprolites not more frequently found in it?

Overall, I'm still not convinced that the gross morphology of the strandflat, as it looks now, dates back to the Triassic. It might, but I do not think that a small number of Triassic saprolites fully demonstrates this. It is true, that the strandflat looks like a weathering surface (line 298), but it also looks like a zone of areal scour (line 69) so it is risky to rely on morphology.

As mentioned before, the landscape probably is a bit of both. We see the Bømlø landscape as an originally Mesozoic etch surface (dissected by joints and faults) , but with a significant Pleistocene polish. However, the Pleistocene erosion has not been complete, since small pockets of saprolite still remain.

We have added a section on the discussion of preservation of landforms or alternatively on successive exhumation of landforms and saprolite outcrops.

See previous comments from Hall, and below on "current erosion level".

However, the main point regarding long-term stability of the coastal area (dismissing km-scale erosion or deposition) stands clearer now (in lines 286-297), and to me this is the most convincing and important finding. In fact, I would recommend that this point be reflected in the title somehow instead of the "Mesozoic landscape".

We prefer to maintain the initial title.

Another point regarding this subject is that the last sentence (line 310) is rather imprecise. What exactly is an "ancient landscape"? To some it might be a landscape just pre-dating the latest glaciations (as often demonstrated using cosmogenic nuclides), to others it means something older, or a surface that has not eroded for a long time. Please be more specific.

Agreed, we changed the sentence.

This leads to a final (minor) comment: the preservation of landscapes in glaciated terrain is often associated with cold-based/non-erosive part of an ice sheet, but the ice-sheets were presumably not cold-based along the coasts of Scandinavia. It would be good to see this discussed a bit – i.e. in few sentences that describes known conditions for landscape preservation under ice sheets.

We have added a section on this. Landscape preservation beneath ice sheets is one aspect of the problem but we also think it is important to remember what Adrian Hall mentioned in his comments to the revised paper; the saprolites we find along the Norwegian coast today represent the current erosion level! Put in other words, the saprolites we sample today will likely be gone after the next glaciation (perhaps 100 ka into the future) and new saprolite outcrops will instead be exposed close to the near shore Mesozoic basins.

Otherwise, I think that the text reads really well, and I look forward to seeing it published.

Thank you

David L Egholm

Reviewer #3 (Remarks to the Author):

I read the revised draft, the reviews, and the responses to the reviews. My review and comments focus on the K-Ar technique and the clay mineralogy of these saprolite samples. The revised text is an improved version all through the text. The authors provided good answers to all questions.

The point sticking me is the presence of smectite in the finest fractions of the Bomlo2 while Ustira and Ivo have some illite or mica in the finest fractions. That smectite must have some fixed K and evidently enough for a useful mass spectrometric analysis (so, yes, you have a more sensitive MS relative to the earlier MS used by many investigators, no need to run an unspiked sample!). The origin of that smectite is a weathering origin? I know the authors go back and say the K-Ar tells the story. The relative similarity of these ages argue for a similar process. Ok, I see it. I advise stressing the fixed K in the smectite and thus a datable phase - given the stress the authors place on the K-Ar ages of these finest fractions. The K value is 0.15 (about what I would guess for smectite). My guess is that a closer look at the XRD pattern for that sample will show mixed layer illite with a small number of illite layers (I am thinking to look for non-integral lower order peak positions or asymmetry of the 17 angstrom peak).

We agree, this is certainly possible, or even likely, and we present that in the SEM images. In fact the illite is certainly quite often interlayered with smectite as shown in the figures. However, we do not think this impacts on the quality of the dating exercise. How the K-bearing is associated with other fine grained (clay) phases should not significantly impact our results. In summary, we think our method is robust considering we perform; i) careful field mapping and sampling to make sure we avoid ambiguous outcrops, ii) extensive XRD and SEM analysis of the samples, and iii) analysis of different clay grain size fractions using centrifuges. These methodologies allow us to isolate authigenic illite clay alone and gives us a good approximation of saprolite, and hence landscape formation.

Good analyses on the LP6.

Line 428: need a "b)" somewhere. Lines 444-445 there are two "e)".

Corrected.

All told, this paper will be a very useful contribution. It will show another use for K-Ar analyses. This study is informative with some very interesting (and rare) field context outcrops. The linkage to Triassic pCO₂ is useful and makes the paper very interesting (terrestrial paleoclimates are difficult to interpret). This study builds knowledge both in geochronology, terrestrial climate proxies, and paleoclimatology.

Thank you.

REVIEWERS' COMMENTS:

Reviewer #2 (Remarks to the Author):

Comments to revised version of "The inheritance of a Mesozoic landscape in western Scandinavia" by Fredin et al.

Overall I think that this third iteration has improved the text. In particular, I like the discussion section and the concluding remarks.

I still think, however, that the point on the strandflat landscape in the bold section (lines 30-32) is stretching the findings of the study too far, as is the title. My concerns regarding the dating of a landscape (as opposed to dating some of the rocks in the landscape, which this study is really doing) thus remains in the new version. On the other hand, the new version of the discussion section is reflecting some of the other possible conclusions that can be inferred from the presence of Triassic saprolites along the coast, so hopefully the readers will read through the full text. I look forward to seeing the study published, and I'm convinced that it will attract wide interest. Congratulations on an interesting and innovative study.